# GAGA: GROUP ANY GAUSSIANS VIA 3D-AWARE MEMORY BANK

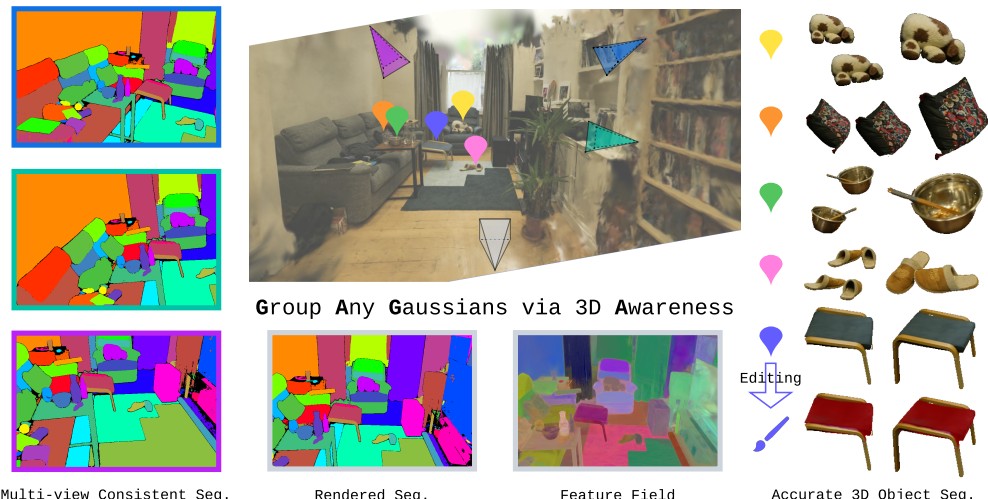

Group Any Gaussians via 3D Awareness

Multi-view Consistent Seg.  Rendered Seg.  Feature Field  Accurate 3D Object Seg.

Figure 1: ***Gaga* groups any Gaussians** in an open-world 3D scene and renders multi-view consistent segmentation (pixels of the same region across views are represented with the same color). By employing a 3D-aware memory bank, we eliminate the label inconsistency that exists in 2D segmentation predicted by foundational models and assign each mask across different views a universal group ID. This enables the process of lifting 2D segmentation to a consistent 3D segmentation. *Gaga* produces accurate 3D object segmentation, achieving high-quality results for downstream applications such as scene manipulation (*e.g.* changing the cushion's color of the footstool to maroon).

## ABSTRACT

We introduce *Gaga*, a framework that reconstructs and segments open-world 3D scenes reconstructed with 3D Gaussians by leveraging inconsistent 2D masks predicted by zero-shot class-agnostic segmentation models. Contrasted to prior 3D scene segmentation approaches that heavily rely on video object tracking, *Gaga* utilizes spatial information provided by 3D Gaussians and effectively associates object masks across diverse camera poses through a novel 3D-aware memory bank. By eliminating the assumption of continuous view changes in training images, *Gaga* demonstrates robustness to variations in camera poses, particularly beneficial for sparsely sampled images, ensuring precise mask label consistency. Furthermore, *Gaga* accommodates 2D segmentation masks from diverse sources and demonstrates robust performance with different open-world zero-shot class-agnostic segmentation models, significantly enhancing its versatility. Extensive qualitative and quantitative evaluations demonstrate that *Gaga* performs favorably against state-of-the-art methods, emphasizing its potential for real-world applications such as scene understanding and manipulation. The source codes will be made available to the public.

## 1 INTRODUCTION

Effective open-world 3D segmentation is essential for scene understanding and manipulation. Despite notable advancements in 2D open-world segmentation techniques, exemplified by Segment Anything (SAM) (Kirillov et al., 2023) and EntitySeg (Qi et al., 2023), extending these methodologies to the realm of 3D encounters the challenge of ensuring consistent mask label assignment

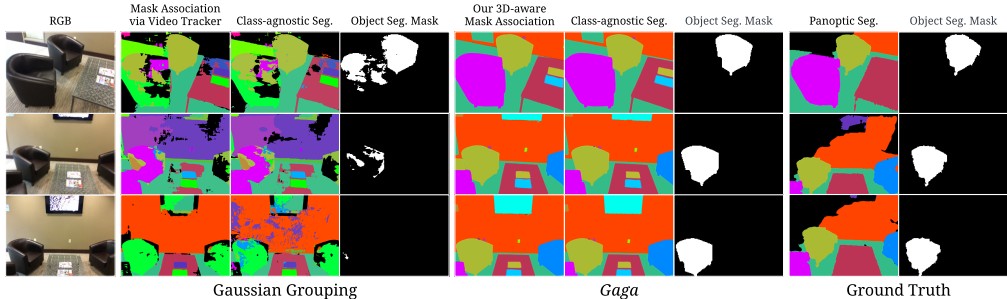

Figure 2: **Comparison of rendered segmentation.** Previous methods (Ye et al., 2024; Dou et al., 2024) adopt an off-the-shelf video object tracker (Cheng et al., 2023a) to for mask association. Results on ScanNet dataset (Dai et al., 2017) show that they frequently misidentify objects, especially when similar objects are present in the scene (*e.g.* the leather sofa), and struggle to handle significant changes in camera perspective. In contrast, *Gaga* integrates 3D information to precisely locate objects and associate 2D masks, leading to multi-view consistent class-agnostic segmentation and precise 3D object segmentation rendering. We adopt the ground truth panoptic segmentation of the ScanNet dataset for comparison as, in this case, it is visually similar to class-agnostic segmentation.

across multi-view images. Specifically, masks of the same object across different views may have different mask label IDs, as the 2D class-agnostic segmentation model independently processes the multi-view images. Naively lifting these inconsistent 2D masks to 3D introduces ambiguity and leads to inferior results in 3D scene segmentation. Hence, we show that assigning each mask a multi-view consistent universal mask ID is crucial before lifting them to 3D. We refer to this task as mask association (Ye et al., 2024).

Prior research efforts (Ye et al., 2024; Dou et al., 2024) leverage the recent advance in 3D reconstruction, e.g., Gaussian Splatting (Kerbl et al., 2023), attempt to solve this task by treating multi-view images as video sequences and adopting an off-the-shelf video object tracking method (Cheng et al., 2023a). Nevertheless, this design relies on the assumption of minimal view changes between multi-view images. This condition may not consistently hold in real-world 3D scenes, especially when the input views are sparse. Consequently, these approaches struggle with similar objects or occluded objects that intermittently disappear and reappear in the sequence, as shown in Fig. 2.

As such, we analyze the fundamental disparity between the 3D mask association and video object-tracking tasks: the latter does not leverage 3D information naturally provided by the 3D scene. Specifically, masks of the same object across different views shall correspond to the same group of 3D Gaussians. Hence, we can assign two masks from different views with the same universal mask ID if there is a large overlap between two groups of 3D Gaussians splatted onto the masks.

In this work, we propose *Gaga*, which groups any 3D Gaussians and renders consistent 3D class-agnostic segmentation across different views. Given a collection of posed RGB images, we first employ Gaussian Splatting (Kerbl et al., 2023) to reconstruct the 3D scene and extract 2D masks using an open-world segmentation method. Subsequently, we iteratively build a 3D-aware memory bank that collects and stores the Gaussians grouped by category. Specifically, for each input view, we project each 2D mask into 3D space using camera parameters and search the memory bank for the category with the largest overlapping with the unprojected mask. Depending on the degree of overlapping, we either assign the mask to an existing category or create a new one. Finally, following the mask association process described above, we can get a set of multi-view consistent 2D segmentation masks.

We use a 16-dimension feature for each Gaussian, i.e., Identity Encoding (Ye et al., 2024). Leveraging the consistent 2D segmentation masks as supervision, identity encoding is trained via differentiable Gaussian rendering to render 2D segmentation feature maps. A linear layer is employed to classify the splatted 2D features into semantic labels for segmentation rendering.

Our approach, *Gaga*, is capable of: 1) synthesizing novel view images of RGB and segmentation with inherent 3D consistency; 2) grouping 3D Gaussians based on their 2D segmentation masks and providing accurate 3D object segmentation for scene manipulation; 3) accommodating any 2D segmentation methods without additional mask pre-processing.

Our contributions are summarized as follows:

- We propose a framework that reconstructs and segments 3D scenes using inconsistent 2D masks generated by open-world segmentation models.
- To resolve the inconsistency of 2D masks across views, we design a 3D-aware memory bank that collects Gaussians of the same semantic group. This memory bank is then employed to align 2D masks across diverse views.
- We show that the proposed method can effectively leverage any 2D segmentation masks, making it readily applicable for synthesizing novel view images and segmentation masks.
- We conduct comprehensive experiments on diverse datasets and challenging scenarios, including sparse input views, to demonstrate the effectiveness of the proposed method both qualitatively and quantitatively.

## 2 RELATED WORK

**Segment and Tracking Anything in 2D.** Segment Anything (SAM) (Kirillov et al., 2023) and EntitySeg (Qi et al., 2023) demonstrate the effectiveness of large-scale training in image segmentation, thus establishing a pivotal foundation for open-world segmentation methods. Subsequent studies (Yang et al., 2023; Cheng et al., 2023b;a) further extend the applicability of SAM to video data by leveraging video object segmentation algorithms to propagate SAM masks. Conversely, acquiring data for training their 3D counterparts poses a challenge, given that existing 3D datasets with annotated segmentation (Straub et al., 2019; Dai et al., 2017) primarily focus on indoor scenarios.

**NeRF-based 3D Segmentation.** Neural Radiance Fields (NeRF) (Mildenhall et al., 2020) model scenes as continuous volumetric functions, learned through neural networks that map 3D coordinates to scene radiance. This approach facilitates the capture of intricate geometric details and the generation of photorealistic renderings, offering novel view synthesis capabilities.

Semantic-NeRF (Zhi et al., 2021) initiates the incorporation of semantic information into NeRFs and enables the generation of semantic masks for novel views. Note that semantic segmentation masks do not face the challenge of ambiguous mask ID across views. Numerous methods expand the scope by introducing instance modeling and matching instance masks relying on existing 3D bounding boxes (Liu et al., 2023b; Fu et al., 2022), resorting to the cost-based linear assignment during training (Siddiqui et al., 2023; Wang et al., 2023) or directly training instance-specific MLPs (Kundu et al., 2022). However, most of these methods are developed based on ground truth segmentation and tailored for scene modeling within specific domains. They often entail high computational costs and lack substantial evidence of their performance in open-world scenarios.

Leveraging SAM's open-world segmentation capability, SA3D (Cen et al., 2023b) endeavors to recover a 3D consistent mask by tracing 2D masks across adjacent views with user guidance. Similarly, Chen et al. (Chen et al., 2023) distill SAM encoder features into 3D and query the decoder. In contrast, *Gaga* achieves multi-view consistency without user intervention, offering segmentation for all objects rather than an instance. Garfield (Kim et al., 2024) focuses more on clustering than segmentation. They densely sample SAM masks and train a scale-conditioned affinity field supervised on the scale of each mask deprojected to 3D.

**Gaussian-based 3D Segmentation.** As an alternative to NeRF and its variants (Mildenhall et al., 2020; Chen et al., 2022; Müller et al., 2022; Tancik et al., 2023), Gaussian Splatting (Kerbl et al., 2023; Chen & Wang, 2024; Wu et al., 2024; Yu et al., 2024) has recently emerged as a powerful approach to reconstruct 3D scenes via real-time radiance field rendering. By representing the scene as 3D Gaussians from posed images, it achieves photorealistic novel view synthesis with high reconstruction quality and efficiency. Additionally, manipulating 3D Gaussians for scene editing is more straightforward compared to NeRF's representation.

SAGA (Cen et al., 2023a) renders a 2D SAM feature map and uses a SAM guidance loss to learn 3D segmentation from the ambiguous 2D masks. Similar to (Cen et al., 2023b), this method requires user input and only provides segmentation for one object at a time. Feature 3DGS (Zhou et al., 2024) distills LSeg (Li et al., 2022) and SAM features to 3D Gaussians and decodes rendered features to obtain segmentation. However, it fails to provide consistent segmentation across views. Gaussian Grouping (Ye et al., 2024) and CoSSegGaussians (Dou et al., 2024) use a video object tracker (Cheng et al., 2023a) to associate masks across different views. However, in scenarios with significant changes in camera poses between frames, such approaches struggle to maintain accuracy.

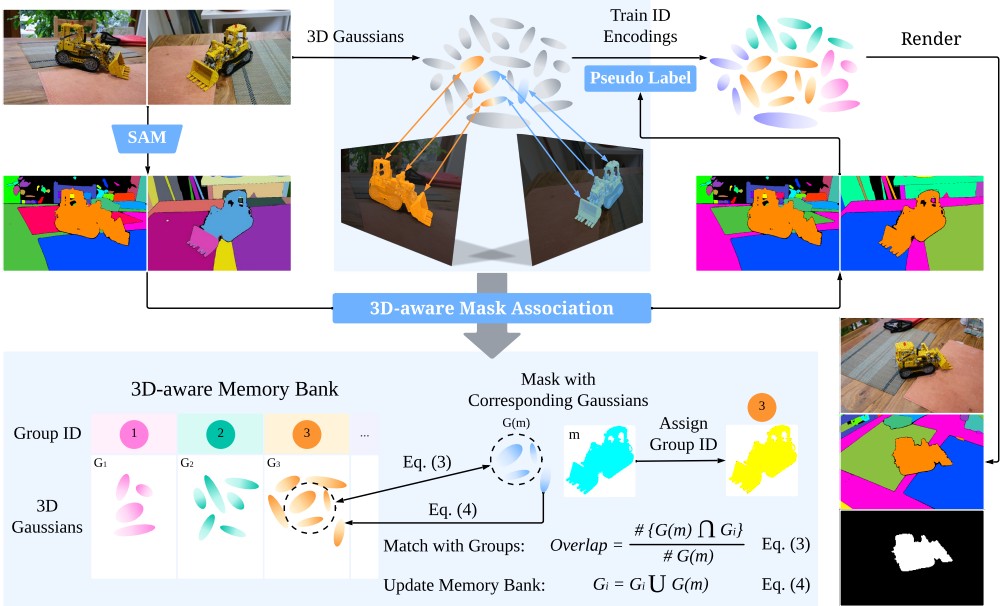

Figure 3: **Overview of *Gaga*.** *Gaga* reconstructs 3D scenes using Gaussian Splatting and adopts any open-world model to generate 2D segmentation masks. To eliminate the 2D mask label inconsistency, we design a mask association process, where a 3D-aware memory bank is employed to assign a consistent group ID across different views to each 2D mask based on the 3D Gaussians projected to that mask (Sec. 3.2). Specifically, we find the corresponding Gaussians projected to 2D mask and assign the mask with the group ID in the memory bank with the maximum overlapped Gaussians (Eq. 3) After 3D-aware mask association process, we use masks with multi-view consistent group IDs as pseudo labels to train an identity encoding on each 3D Gaussian for segmentation rendering.

# 3 PROPOSED METHOD

## 3.1 PRELIMINARIES

**Gaussian Splatting.** Recently, Gaussian Splatting (Kerbl et al., 2023) has significantly advanced the 3D representation field by combining the benefits of implicit and explicit 3D representations. Specifically, a 3D scene is parameterized as a set of 3D Gaussians $\{G_i\}$. Each Gaussian $G_i = \{p_i, s_i, q_i, \alpha_i, c_i\}$ is defined by its position $p_i = \{x, y, z\} \in \mathbb{R}^3$, scale $s_i \in \mathbb{R}^3$, orientation $q \in \mathbb{R}^4$, opacity $\alpha \in \mathbb{R}$ and color features $c$ encoded by spherical harmonics (SH) coefficients.

Gaussian Splatting employs the splatting rendering pipeline, wherein 3D Gaussians are projected onto the 2D image space using the world-to-frame transformation matrix corresponding to each camera pose. Gaussians projected to the same coordinates $(x, y)$ (represented as $i \in N$) are blended in depth order and weighted by their opacity $\alpha$ to produce the color $c_{x,y}$ of each pixel:

$$c_{x,y} = \sum_{i \in N} c_i \alpha_i \prod_{j=1}^{i-1} (1 - \alpha_j). \tag{1}$$

**Identity Encoding.** Identity encoding (Ye et al., 2024) aims to assign a universal label to each 3D Gaussian for segmentation rendering. It is a 16-dimensional feature attached to each Gaussian, which is subsequently decoded to a segmentation mask ID through a combination of linear and SoftMax layers. The resulting mask ID can then be supervised using 2D segmentation masks.

$$m_{x,y} = \arg\max\{L(\sum_{i \in N} e_i \alpha_i \prod_{j=1}^{i-1} (1 - \alpha_j))\}. \tag{2}$$

## 3.2 GROUP ANY GAUSSIANS VIA 3D-AWARE MEMORY BANK

Given a set of posed images, we aim to reconstruct a 3D scene with semantic labels for segmentation rendering. To this end, we first leverage Gaussian Splatting for scene reconstruction. We then employ open-world 2D segmentation methods such as SAM (Kirillov et al., 2023) or EntitySeg (Qi et al., 2023) to predict class-agnostic segmentation for each input image.

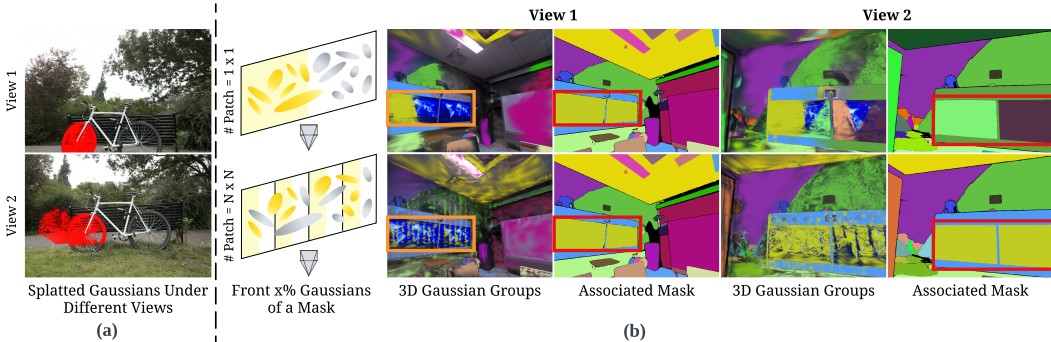

Figure 4: **Illustrations of finding corresponding Gaussians.** **(a) Motivation to choose front** $x\%$ **of Gaussians.** We select $x\%$ of Gaussians closest to the camera frame as many Gaussians splatted to mask in view 1 represent objects from behind, as shown in view 2. **(b) Significance of mask partition.** We color the Gaussians in the 3D-aware memory bank based on their groups, displayed as `3D Gaussian Groups` in columns 2, and 4. When images aren't partitioned (row 1), the front $x\%$ of Gaussians concentrate in a confined area, as shown in the orange rectangle, also visualized as the yellow Gaussians in column 1. These Gaussians fail to accurately represent the mask's shape, resulting in mismatched masks (columns 3, 5), as shown in the red rectangle. After dividing each mask into N×N patches (row 2) (figure column 1 use 4×1 for illustration), the selected front Gaussians can better represent the shape of a 3D object.

However, because the segmentation model processes each input image independently, the resulting masks are not naturally multi-view consistent. To resolve this issue, (Ye et al., 2024; Dou et al., 2024) assume that nearby input views are similar and apply a video tracker to associate inconsistent 2D masks of different views. Yet, this assumption may not hold for all 3D scenes, especially when the input views are sparse, as demonstrated in Fig. 2.

*Gaga* is inspired by the fundamental disparity between the task of mask association across multiple views and tracking objects in a video: the incorporation of 3D information. To reliably generate consistent masks across different views, we propose a method that leverages 3D information without relying on any assumptions about the input images. Our key insight is that masks belonging to the same object in different views shall correspond to the same Gaussians in the 3D space. Consequently, these Gaussians should be grouped together and assigned an identical group ID.

**Corresponding Gaussians of Mask.** Based on this intuition, we first associate each 2D segmentation mask with its corresponding 3D Gaussians. Specifically, we splat all 3D Gaussians onto the camera frame, given the camera pose of each input image. Subsequently, for each mask within the image, we identify 3D Gaussians whose centers are projected within that mask. Those Gaussians should be identified as representatives of the mask in 3D and can be used as guidance for associating masks from different views.

Notably, segmentation masks typically describe the shape of foreground objects under the current camera pose. However, as Fig. 4 (a) shows, a significant proportion of Gaussians do not contribute to the pixels in the 2D segmentation mask, as they represent objects in the back of the 3D scene. To address this, we experiment with different solutions and find that selecting the front $x$ percentage of 3D Gaussians closest to the camera frame as the corresponding Gaussians of the mask works best. Here, $x$ serves as a hyperparameter that can be adjusted based on the nature of the current 3D scene.

As shown in Fig. 4 (b), some masks in 2D represent a large object in 3D, i.e., the blue screen in the image. Selecting corresponding Gaussians based on the entire mask will inaccurately represent the shape of the 3D object, as the selected front Gaussians often concentrate in a confined area (see the yellow Gaussians in `Front x% Gaussians of a Mask`). Subsequently, when the camera pose changes, the front $x\%$ of Gaussians of the same mask may land in a different confined area (see Fig. 4 (b) View 2 row 1 `3D Gaussian Group`) and have no or little overlap with the Gaussian Group in the memory bank, fail to associate masks across different camera poses. To resolve this issue, we propose a strategy wherein we partition an input image into patches and calculate corresponding Gaussians within each patch, as shown in Fig. 4 (b) row 2. Specifically, we begin by dividing the image into $N \times N$ patches. Subsequently, we identify the collection of top $x\%$ of 3D Gaussians closest to the camera frame within each patch to be the corresponding Gaussians of

mask $m$, denoted as $\mathcal{G}(m)$. This simple strategy effectively improves the consistency of associated masks across different views.

**3D-aware Memory Bank.** Next, to collect and categorize 3D Gaussians into groups and use them to associate masks across different views, we introduce a 3D-aware Memory Bank (see Fig. 3). Given a set of images, we initialize the 3D-aware Memory Bank by storing the corresponding Gaussians of each mask in the first image into an individual group and labeling each mask with a group ID the same as its mask label. For each 2D mask of the subsequent image, we first determine its corresponding Gaussians as outlined above. We then either assign these Gaussians to an existing group within the memory bank or establish a new one if they do not share similarities with existing categories in the memory bank. Details of this assignment process are elaborated in the following.

**Group ID Assignment via Gaussian Overlap.** To assign each mask a group ID, we aim to find if the current mask has a significant amount of overlapped Gaussians with any groups in the memory bank. We define the similarity between two sets of 3D Gaussians based on their shared 3D Gaussian ratio. Specifically, given the 3D Gaussians corresponding to a 2D mask $m$ (denoted as $\mathcal{G}(m)$ as described above) and the Gaussians of category $i$ (denoted as $\mathcal{G}_i$) in the memory bank, we identify their shared Gaussians as $\mathcal{G}(m) \cap \mathcal{G}_i$ (i.e., Gaussians of the same indices), we then compute the overlap as the ratio of number of shared Gaussians to number of all Gaussians within mask $m$:

$$Overlap(m, i) = \frac{\#(\mathcal{G}(m) \cap \mathcal{G}_i)}{\#\mathcal{G}(m)}. \tag{3}$$

Suppose category $i$ has the highest overlap with mask $m$ among all categories in the memory bank, and this overlap value is above a threshold. In that case, we assign the group ID of mask $m$ as $i$ and add the non-overlapped Gaussians in the $i_{th}$ category.

$$\mathcal{G}_i = \mathcal{G}_i \cup \mathcal{G}(m). \tag{4}$$

We establish a new group ID $j$ if none of the existing groups contains an overlap with mask $m$ above the overlap threshold. We add $\mathcal{G}(m)$ into this new category in the Gaussian memory bank and assign mask $m$ with the new group ID $j$. Note that we ensure that each Gaussian is added to only one group in the memory bank by recording all the Gaussian indices already in the memory bank.

### 3.3 3D Segmentation Rendering and Downstream Application.

After the group ID assigning process, masks projected by the same group of Gaussians are supposed to have the same group ID across different views. Similar to (Ye et al., 2024), we learn a 16-dimension identity encoding feature for each Gaussian. Like RGB rendering, identity encoding renders a 2D identity encoding feature map given a camera pose. Then, a linear layer and a SoftMax function are employed to predict a semantic label for each pixel in the rendered identity encoding feature map, resulting in segmentation rendering. We use those associated masks as pseudo-labels to train the identity encoding.

Our segmentation-aware 3D Gaussians can be readily used for various downstream applications. For instance, we can render segmentation masks of novel views with consistent semantic labels for the same object across different camera poses. Gaussians can also be selected by their identity encoding for scene editing tasks, including removal, color-changing, position translation, etc., as demonstrated in Sec. 4.6.

## 4 Experiments

### 4.1 Experimental Setup

**Datasets.** We experiment with various datasets across diverse scenarios to demonstrate the performance of *Gaga*. For quantitative comparison, we use a scene understanding dataset LERF-Mask (Ye et al., 2024), along with two indoor scene datasets: Replica (Straub et al., 2019) and ScanNet (Dai et al., 2017). Additionally, we showcase the robustness of *Gaga* against variations in training image quantity by sparsely sampling the Replica dataset. We present visual comparison results on the commonly used scene reconstruction dataset, MipNeRF 360 (Barron et al., 2021). The quantitative and qualitative results are conducted on the test set, i.e., novel view synthesis results. Details about datasets can be found in the supplementary material.

**Evaluation Metrics.** Similarly to prior work (Ye et al., 2024), mIoU and boundary IoU (mBIoU) are used to evaluate the LERF-Mask dataset. We use ground-truth panoptic segmentation to evaluate the Replica and ScanNet datasets and disregard class information. To handle differences between

Table 1: **Quantitative results for open-vocabulary 3D query tasks on LERF-Mask dataset.** *Gaga* outperforms previous approaches, showcasing favorable performance in terms of mIoU and BIoU with both segmentation models. * denotes the results are reported in (Ye et al., 2024).

| Model | 2D Seg. Method | mIoU(%) | mBIoU(%) |
|---|---|---|---|
| LERF (Kerr et al., 2023)* | / | 37.17 | 29.30 |
| LEGaussians (Shi et al., 2024) | / | 36.26 | 28.90 |
| LangSplat (Qin et al., 2024) | / | 61.24 | 56.07 |
| Gaussian Grouping (Ye et al., 2024) | EntitySeg | 54.10 | 50.90 |
| *Gaga* (Ours) | | 62.44 | 60.28 |
| Gaussian Grouping (Ye et al., 2024)* | SAM | 72.79 | 67.58 |
| *Gaga* (Ours) | | **74.71** | **72.19** |

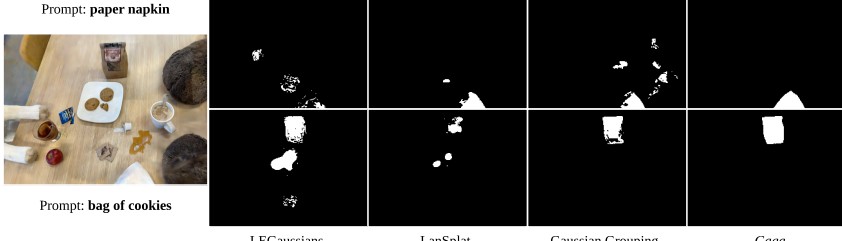

Prompt: **paper napkin**

Prompt: **bag of cookies**

LEGaussians    LanSplat    Gaussian Grouping    *Gaga*

Figure 5: **Visual comparison on LERF-Mask dataset.** Our rendered segmentation exhibits fewer artifacts and delivers more accurate segmentation results than both prior 3D class-agnostic segmentation works and language embedding works.

predicted and ground truth mask labels, we calculate the best linear assignment based on IoU. Moreover, with IoU = 0.5 as the criterion, we report precision and recall to further evaluate the accuracy of predicted masks.

**Implementation Details.** We use SAM (Kirillov et al., 2023) and EntitySeg (Qi et al., 2023) with the Hornet-L backbone to obtain open-world 2D segmentation. We preprocess the generated raw masks following the method outlined in (Qi et al., 2023), prioritizing those with higher confidence scores by ranking them accordingly. Masks with confidence scores below 0.5 are discarded. For all experiments, we train vanilla Gaussian Splatting (Kerbl et al., 2023) for 30K iterations and train the identity encoding for 10K iterations with all other parameters frozen. We train Gaussian Grouping (Ye et al., 2024) for 40K iterations for fair comparisons. We choose the front 20% 3D Gaussians that are closest to the camera frames to be the corresponding Gaussians of a mask. We set the overlap threshold for declaring a new group ID as 0.1. We divide the image into 32×32 patches. Ablation studies of these parameter choices can be found in the supplementary material, showing that our method is robust to parameters selection.

### 4.2 OPEN-VOCABULARY 3D QUERY ON LERF-MASK DATASET

We compare our method with 4 state-of-the-art methods on 3D scene understanding: LERF (Kerr et al., 2023), LEGaussians (Shi et al., 2024), LangSplat (Qin et al., 2024) and Gaussian Grouping (Ye et al., 2024). The first three methods focus on CLIP feature embedding. We calculate the relevancy between rendered CLIP features and query text features following their official implementation. Since SAM (Kirillov et al., 2023) and EntitySeg (Qi et al., 2023) do not support language prompts, both *Gaga* and Gaussian Grouping adopt Grounding DINO (Liu et al., 2023a) to identify the mask ID in a 2D image and pick the corresponding 3D mask. Tab. 1 illustrates that *Gaga* achieves superior results in mIoU and mBIoU compared to previous methods, especially when utilizing SAM as 2D segmentation method. Visualization results of 3D query tasks with prompts "paper napkin" and "bag of cookies" further demonstrate the advancement of *Gaga*.

### 4.3 3D SEGMENTATION ON REPLICA AND SCANNET DATASETS

Tab. 2 presents the quantitative comparison results on Replica and ScanNet datasets. *Gaga* exhibits better performance on both datasets regardless of the 2D segmentation model used, showcasing its stability across different datasets and models and consistently achieving high performance. Qualitative results are shown in Fig. 6. Rows 1-2 depict the visualizations from the Replica dataset, while rows 3-5 showcase results from the ScanNet dataset. Gaussian Grouping (Ye et al., 2024) frequently assigns different mask IDs to the same object, resulting in inconsistent mask colors and empty regions. Row 4 illustrates that Gaussian Grouping struggles to distinguish similar objects, whereas our proposed *Gaga* accurately identifies each object by leveraging 3D information.

Table 2: **Quantitative results on Replica and ScanNet datasets.** *Gaga* performs well with both 2D segmentation methods on two datasets. The performance of Gaussian Grouping varies significantly with different 2D segmentation methods, whereas *Gaga* consistently delivers stable performance.

| Model | 2D Seg. Method | Replica | | | ScanNet | | |
|---|---|---|---|---|---|---|---|
| | | IoU(%) | Precision(%) | Recall(%) | IoU(%) | Precision(%) | Recall(%) |
| Gaussian Grouping (Ye et al., 2024) | EntitySeg | 35.90 | 14.07 | 31.57 | 39.54 | 6.88 | 36.56 |
| *Gaga* (Ours) | | 41.08 | **63.06** | 46.14 | 42.56 | **33.89** | **47.63** |
| Gaussian Grouping (Ye et al., 2024) | SAM | 21.76 | 25.00 | 19.72 | 34.24 | 18.70 | 32.61 |
| *Gaga* (Ours) | | **46.50** | 41.52 | **52.50** | **44.87** | 18.61 | 45.94 |

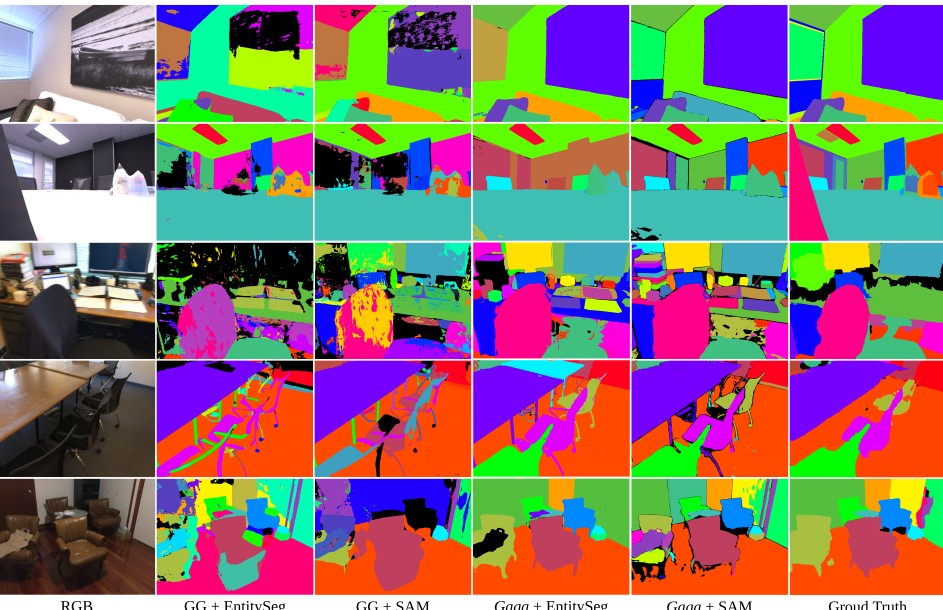

| RGB | GG + EntitySeg | GG + SAM | *Gaga* + EntitySeg | *Gaga* + SAM | Groud Truth |

Figure 6: **Qualitative results on Replica and ScanNet datasets.** *Gaga* provides high-quality segmentation masks that are more similar to the ground truth. Gaussian Grouping (noted as GG) often covers the same object with different masks (rows 1, 3, 6), creating large empty regions (rows 1-3), and misidentifying similar instances (rows 4, 5).

## 4.4 3D SEGMENTATION WITH LIMITED DATA ON REPLICA DATASET

To demonstrate the robustness of *Gaga* against changes in training image quantity, we sparsely sample the Replica training set with ratios of 0.3, 0.2, 0.1, and 0.05. As depicted in Tab. 3, *Gaga* consistently exhibits superior performance in terms of IoU, with approximately a 10% advantage using EntitySeg and a 20% advantage using SAM. Remarkably, when utilizing SAM, *Gaga* surpasses fully trained Gaussian Grouping with just 5% of the training data (22.79% vs. 21.76%). We also compute the IoU drop compared to using all training images, as follows:

$$IoU\ Drop(x\%) = \frac{IoU(100\%) - IoU(x\%)}{IoU(100\%)}, \tag{5}$$

Table 3: **Quantitative results on Replica dataset with limited training data.** *Gaga* consistently outperforms Gaussian Grouping with both 2D segmentation methods. The percentage of IoU drop indicates that *Gaga* exhibits greater robustness against reductions in training data.

| Model | 2D Seg. Method Training Data | EntitySeg | | SAM | |
|---|---|---|---|---|---|
| | | IoU(%) ↑ | IoU Drop(%) ↓ | IoU(%) ↑ | IoU Drop(%) ↓ |
| Gaussian Grouping (Ye et al., 2024) | 30% | 28.42 | 20.85 | 17.02 | 21.78 |
| *Gaga* (Ours) | | 37.98 | 7.57 | 41.79 | 10.11 |
| Gaussian Grouping (Ye et al., 2024) | 20% | 24.56 | 31.35 | 16.02 | 26.38 |
| *Gaga* (Ours) | | 37.25 | 9.33 | 40.27 | 13.40 |
| Gaussian Grouping (Ye et al., 2024) | 10% | 20.62 | 42.56 | 13.97 | 35.78 |
| *Gaga* (Ours) | | 31.93 | 22.27 | 35.61 | 23.40 |
| Gaussian Grouping (Ye et al., 2024) | 5% | 10.00 | 72.15 | 6.77 | 68.87 |
| *Gaga* (Ours) | | 20.59 | 49.88 | 22.79 | 50.98 |

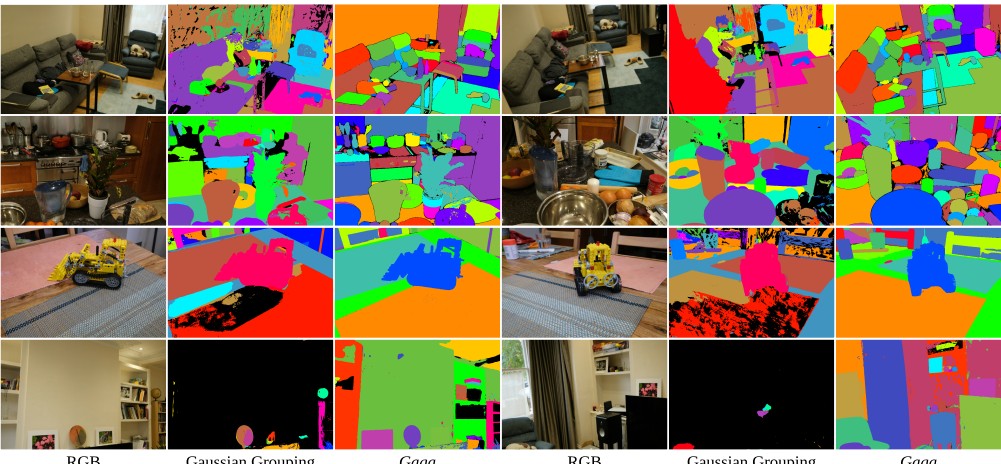

Figure 7: **Qualitative results on Replica dataset with limited training data**. The visualizations depict samples when using only 5% of the training data. Even with limited data, *Gaga* consistently produces high-quality segmentation. In contrast, Gaussian Grouping struggles to track objects accurately and leaves significant empty regions.

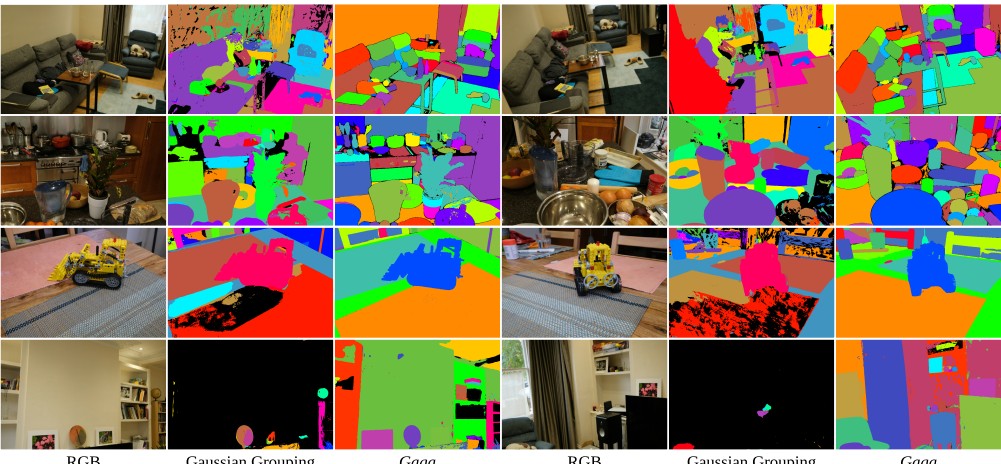

Figure 8: **Qualitative results on MipNeRF 360 dataset.** *Gaga* provides superior segmentation with finer details (row 1, 2), fewer artifacts and empty regions (row 1, 3, 4), and more consistent object segmentation across different views (wall in row 1, tablecloth in row 3).

where $IoU(x\%)$ denotes the IoU achieved when $x\%$ of the training data is used. Compared to Gaussian Grouping, *Gaga* exhibits less sensitivity to decreases in the number of training images, as evidenced by smaller values in IoU drop. Visualization results are shown in Fig. 7. With just 5% of the training data, *Gaga* can still deliver accurate segmentation masks, whereas Gaussian Grouping fails to provide masks for a significant portion of objects due to inaccurate tracking.

### 4.5 3D SEGMENTATION ON MIPNERF 360 DATASET

We further showcase the performance of *Gaga* on the diverse MipNeRF 360 dataset. We provide visualization comparison with Gaussian Grouping (Ye et al., 2024) in Fig. 8, using SAM (Kirillov et al., 2023) (row 1, 2) and EntitySeg (Qi et al., 2023) (row 3, 4). We display two images for each scene to assess the consistency across different views. *Gaga* offers more detailed segmentation, while segmentation masks generated by Gaussian Grouping exhibit severe artifacts and empty regions. Additionally, inconsistency across two views exists in the rendering results of Gaussian Grouping, as shown in row 1, 3.

### 4.6 APPLICATION: SCENE MANIPULATION

*Gaga* achieves high-quality and multi-view consistent 3D segmentation, beneficial for tasks like scene manipulation. We demonstrate the effectiveness of this application on the MipNeRF 360 (Barron et al., 2021) and Replica (Straub et al., 2019) datasets (see Fig. 9). In the left image, we change the color of the footstool's cushion and remove the stuffed animal on the armchair. *Gaga* accurately identifies 3D Gaussians representing the cushion, while Gaussian Grouping fails, coloring the entire footstool maroon along with part of the sofa and some floating Gaussians. *Gaga* also effectively groups and removes the entire part of the stuffed animal on the sofa with minimal artifacts, whereas Gaussian Grouping leaves many floating Gaussians. Similar results are observed in experiments involving the position shifting of a chair in the right image.

### 4.7 ABLATION STUDY ON MASK ASSOCIATION METHOD

We conduct ablation studies to evaluate the effectiveness of the proposed mask association method on the Replica dataset. The baselines for comparison include: 1) *w/o. Mask Association*: Lifting

Task: Change the cushion's color of 🪑 to maroon, remove 🐕     Task: Move 🪑 closer to the window

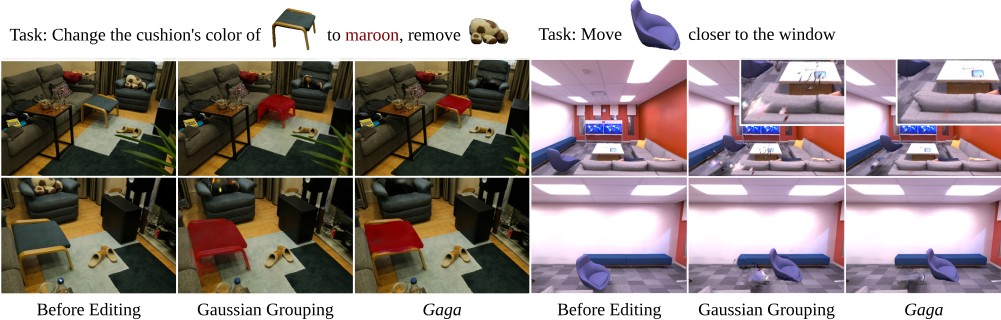

Before Editing    Gaussian Grouping    *Gaga*     Before Editing    Gaussian Grouping    *Gaga*

Figure 9: **Scene manipulation results on MipNeRF 360 and Replica dataset.** *Gaga* accurately identifies the cushion part of the footstool, whereas Gaussian Grouping colors it entirely. For object removal and translation tasks, *Gaga* generates more precise 3D entities with fewer artifacts, resulting in better visual performance.

Table 4: **Ablation study on different mask association methods.** Our mask association method with the 3D-aware memory bank surpasses the previous video tracker baseline on both IoU, Precision, and Recall.

| Baseline | IoU (%) | Precision (%) | Recall (%) |
|---|---|---|---|
| SAM (Upper Bound) | 60.89 | 57.07 | 67.16 |
| w/o. Mask Association | 8.81 | 3.19 | 2.16 |
| Video Tracker (Ye et al., 2024) | 21.76 | 25.00 | 19.72 |
| Memory Bank (w. All Gaussians) (Ours) | 42.26 | 40.19 | 45.95 |
| Memory Bank (w/o. Mask Partition) (Ours) | 46.08 | 27.88 | 50.67 |
| Memory Bank (Depth) (Ours) | 46.09 | 36.30 | 51.25 |
| Memory Bank (Ours) | **46.50** | **41.52** | **52.50** |

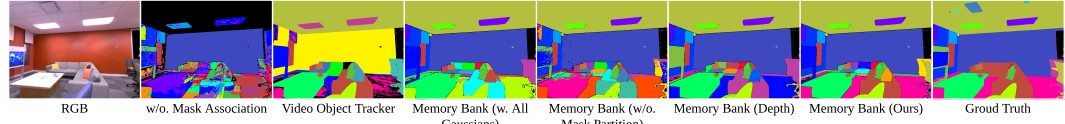

RGB    w/o. Mask Association    Video Object Tracker    Memory Bank (w. All Gaussians)    Memory Bank (w/o. Mask Partition)    Memory Bank (Depth)    Memory Bank (Ours)    Ground Truth

Figure 10: **Visual comparison of different mask association methods.** *Gaga* with 3D-aware memory bank achieves a superior visual quality and closer to the ground truth. Notice that the Video Tracker baseline mislabels the wall and floor, Memory Bank (w. All Gaussians) and Memory Bank (Depth) baselines mislabel the floor, Memory Bank (w/o. Mask Partition) baseline creates artifacts on the table,

inconsistent 2D masks to 3D. 2) *Video Tracker*: (Ye et al., 2024) is employed as a representative method. 3) *Memory Bank (w. All Gaussians)*: same as *Gaga* except that it selects all Gaussians splatted to the mask as its corresponding Gaussians. 4) *Memory Bank (w/o. Mask Partition)*: same as *Gaga* except that it does not partition the image and masks into patches. 5) Memory Bank (Depth): same as *Gaga* except that it uses rendered depth to locate corresponding Gaussians. 6) *Memory Bank*: i.e., *Gaga*. We also add a baseline *SAM (Upper Bound)*, which uses SAM to process rendered RGBs from Gaussian Splatting and evaluate on each single frame without considering multi-view consistency. This baseline can serve as an upper bound to show the inherent difference between class-agnostic and panoptic segmentation.

Quantitative results in Tab. 4 indicate that *Gaga* with the 3D-aware memory bank achieves superior performance with a 24.74%, 16.52%, and 32.78% improvement on IoU, precision on recall, respectively, compared to the previous method with video tracker. Comparison with other Memory Bank baselines demonstrates the effectiveness of our well-designed process for finding corresponding Gaussians of each mask. We also show the visual comparison in Fig. 10.

## 5 CONCLUSIONS

We introduce *Gaga*, a framework that reconstructs and segments open-world 3D scenes using inconsistent 2D masks predicted by zero-shot segmentation models. *Gaga* employs a 3D-aware memory bank to store the indices of pre-trained 3D Gaussians and establishes mask association across different views by identifying the overlap between Gaussians projected to each mask. Results on various datasets demonstrate that *Gaga* outperforms previous methods with superior segmentation accuracy, multi-view consistency, and reduced artifacts. Additionally, application in scene manipulation highlights *Gaga*'s high segmentation accuracy and practical utility.

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

SUPPLEMENTARY MATERIAL

In this supplementary document, we provide further experimental results, including a qualitative comparison with GARField (Kim et al., 2024), more results on scene manipulation and sparse view setting in Sec. B. We then delve into more experimental details of the datasets, metrics and implementation in Sec. C. More ablation studies are shown in Sec. D and limitations are discussed in Sec. E.

## A SUPPLEMENTARY VIDEO

Please watch the supplementary demo video for a comprehensive introduction and visual comparison between our method *Gaga* and the current state-of-the-art methods. The video features additional qualitative comparisons and an animation illustration of *Gaga*.

## B SUPPLEMENTARY EXPERIMENTAL RESULTS

### B.1 ADDITIONAL RESULTS COMPARED WITH GARFIELD

We provide comparison results with GARField in Fig. 11. GARField follows a hierarchical grouping pipeline. It extracts densely sampled segmentation masks from SAM (Kirillov et al., 2023) and trains a feature field using contrastive loss for grouping. If two rays fall into the same SAM mask, their features will be pulled together. Otherwise, features are pushed apart.

We use the default setting to train GARField. For a fair comparison, *Gaga* also uses the 2D segmentation masks provided by SAM. Visualization results show that *Gaga* provides segmentation masks with better quality and multi-view consistency. Whereas GARField does not provide multi-view consistent segmentation, and they also have inferior RGB rendering results.

After training, GARField employs a hierarchical grouping pipeline to cluster each pixel into groups and generate segmentation masks. This hierarchical structure comprises 41 levels, and it takes approximately 20 minutes to output a segmentation mask for a single image. In contrast, *Gaga* renders a segmentation mask in under 0.5 seconds.

### B.2 ADDITIONAL RESULTS ON SCENE MANIPULATION

*Gaga* can accurately segment the Gaussians of a 3D object and edit their properties. Using a pretrained 3D Gaussian model with identity encoding, we employ the classifier trained with identity encoding to predict mask labels for each 3D Gaussian. Subsequently, we select 3D Gaussians shar-

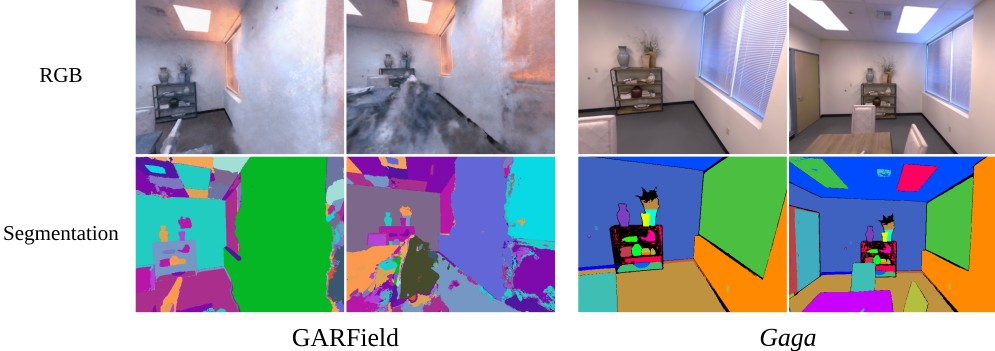

GARField          *Gaga*

Figure 11: **Qualitative comparison with GARField on Replica dataset.** *Gaga* renders higher-quality RGB and segmentation masks in significantly less time. It's worth noting that in the segmentation masks generated by GARField, the same colors are used multiple times for different masks, meaning one mask label may contain multiple groups representing different 3D instances. This is because, essentially, GARField performs a clustering task rather than a segmentation task.

Task: Change the color of flowerpot to cyan, duplicate the glass jar    Task: Change the color duck to blue, remove the red toy chair

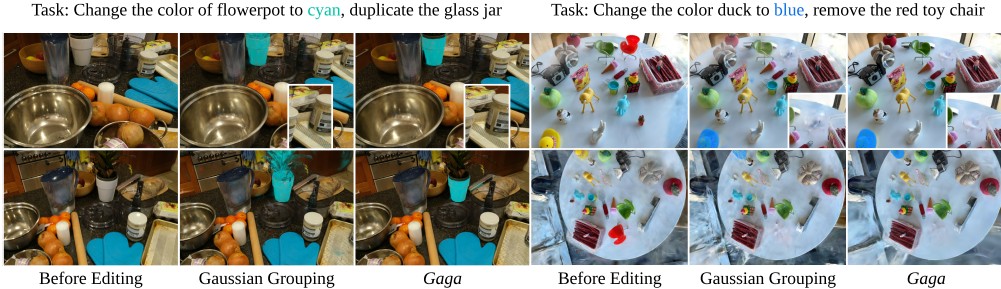

Before Editing        Gaussian Grouping        *Gaga*        Before Editing        Gaussian Grouping        *Gaga*

Figure 12: **Scene manipulation results on MipNeRF 360 and LERF-Mask dataset.** *Gaga* accurately identifies the flowerpot without affecting the color of the plant. Notice that Gaussian Grouping (Ye et al., 2024) creates a cyan region on the wooden door behind. For the object removal and duplication tasks, *Gaga* can also provide more accurate results with fewer artifacts.

ing the same mask label as the target object and edit their properties for tasks like object coloring, removal, and position translation.

We provide additional results for the downstream scene manipulation task to further demonstrate the prospect of applying *Gaga* to real-world scenarios. On the "counter" scene of the MipNeRF 360 dataset (Barron et al., 2021), we change the color of the flowerpot to cyan and duplicate the glass jar. Gaussian Grouping (Ye et al., 2024) can not differentiate the plant and flowerpot, whereas *Gaga* generates a more accurate segmentation mask. Additionally, *Gaga* produces a clearer boundary and avoids artifacts on the iron tray when duplicating the glass jar.

In the "figurines" scene of the LERF-Mask dataset (Ye et al., 2024), we transform the yellow duck to blue and remove the red toy chair. *Gaga* precisely changes only the duck's color without affecting other objects, and achieves a more thorough removal of the red toy chair.

### B.3    ADDITIONAL RESULTS ON SPARSELY SAMPLED REPLICA DATASET

We provide additional qualitative results for the experiment on the sparsely sampled replica dataset in Fig. 13. As the number of training images decreases, Gaussian Grouping produces more empty regions, *e.g.* the sofa, due to difficulties in accurate tracking under sparse views. Whereas *Gaga* exhibits a more robust performance against reductions in the number of images.

### B.4    ADDITIONAL RESULTS ON OUTDOOR SCENE IN MIPNERF 360 DATASET

As most of our results focus on indoor scenarios, we present visual results for outdoor scenes from the MipNeRF 360 Dataset in Fig. 14. To highlight the robustness of *Gaga* in data-limited conditions, this experiment uses a sparsely sampled dataset, utilizing only one-third of the images to reconstruct the scene. The results demonstrate that *Gaga* achieves superior performance in outdoor scenarios as well, producing fewer empty regions and artifacts.

### B.5    ADDITIONAL RESULTS ON TANKS AND TEMPLES DATASET

To further evaluate *Gaga*'s performance on large-scale outdoor scenes and the robustness of its hyperparameter selection across various scenarios, we tested our method on outdoor scenes from the Tanks and Temples (Knapitsch et al., 2017) dataset. Using processed data from (Fan et al., 2024), we reconstructed scenes and rendered segmentation maps with 24 input images. The rendered RGB images, segmentation maps, and feature maps are presented in Fig. 15. These results highlight *Gaga*'s ability to effectively handle large-scale outdoor scenes, demonstrating its robustness.

### B.6    ADDITIONAL RESULTS COMPARED WITH LINEAR ASSIGNMENT ASSOCIATION
####        METHOD

Panoptic Lifting (Siddiqui et al., 2023) uses a cost-based linear assignment method for mask association. We find that this method leads to inferior performance on segmentation and is not suitable for

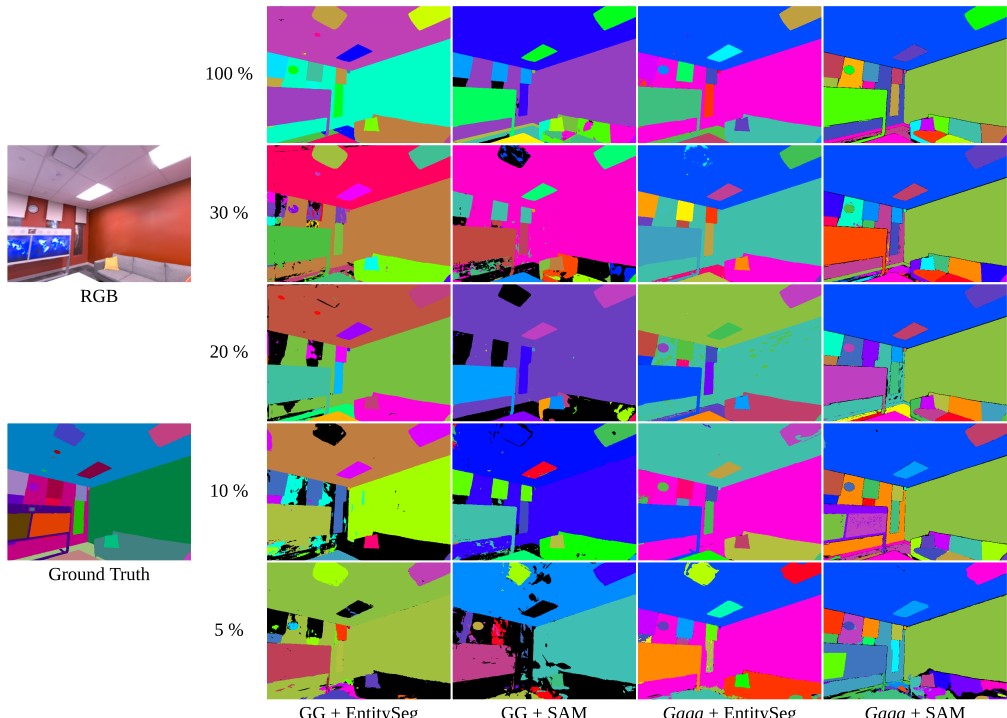

Figure 13: **Qualitative results on the sparsely sampled replica dataset.** We showcase the novel view synthesis segmentation rendering results provided by Gaussian Grouping and *Gaga* as the percentage of training images employed decreases from 100% to 5%. Gaussian Grouping cannot correctly track the sofa under sparse views and fails to differentiate ceiling and wall, whereas *Gaga* consistently provides high-quality segmentation results.

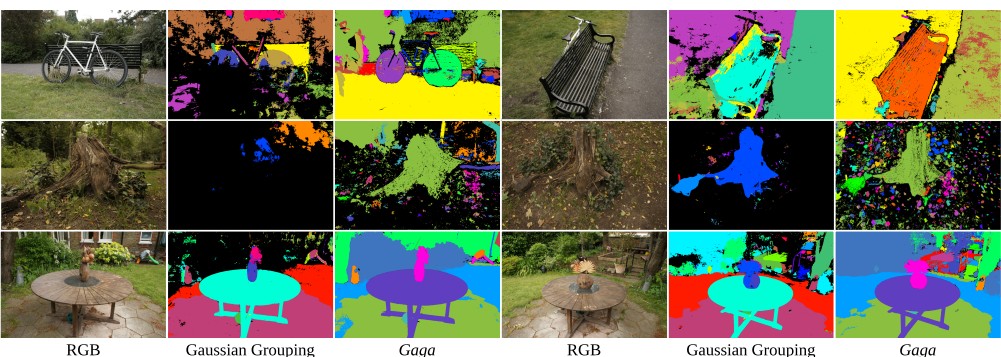

Figure 14: **Visual results on the outdoor scenes.** *Gaga* demonstrates better performance on outdoor scenarios, compared with the previous method Gaussian Grouping Ye et al. (2024). *Gaga* can successfully segment delicate details, *e.g.* the tiny flowers on row 2, while Ye et al. (2024) creates artifacts and large empty regions due to mask association failure.

handling open-world class-agnostic segmentation. Fig. 4 of (Ye et al., 2024) also shows a similar finding.

## B.7 ADDITIONAL RESULTS COMPARED WITH OMNISEG3D

We provide a visual comparison with OmniSeg3D (Ying et al., 2024), which takes multi-view class-agnostic 2D segmentations as input and outputs a consistent 3D feature field using a hierarchical contrastive learning framework. Since OmniSeg3D produces feature renderings but only supports object

| RGB | Segmentation | Feature Map | RGB | Segmentation | Feature Map |
|---|---|---|---|---|---|

Figure 15: **Visual results on Tanks and temples dataset.** We present *Gaga*'s rendered RGB images, segmentation maps, and feature maps on outdoor scenes from the Tanks and Temples dataset. *Gaga* consistently delivers multi-view coherent results on these large-scale outdoor environments, showcasing its robustness across diverse scenarios.

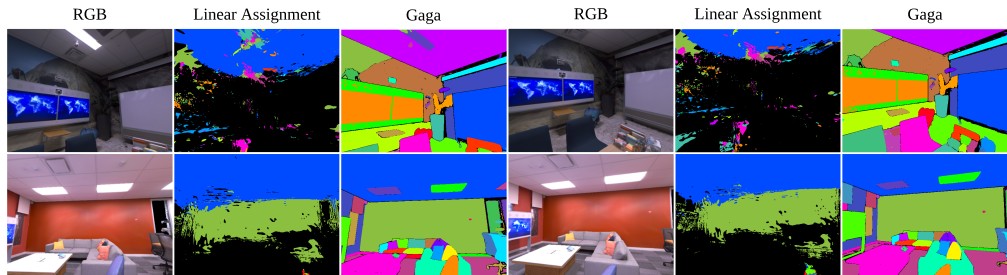

| RGB | Linear Assignment | Gaga | RGB | Linear Assignment | Gaga |
|---|---|---|---|---|---|

Figure 16: **Visual comparison with linear assignment mask association method.** Cost-based linear assignment (Siddiqui et al., 2023) leads to inferior performance on open-world class-agnostic segmentation.

segmentation interactively, we employ the HDBSCAN clustering algorithm to group the rendered features into clusters and generate global segmentation maps. As shown in Fig. 17, the segmentation renderings from OmniSeg3D lack multi-view consistency. Additionally, directly clustering features attached to each Gaussian in 3D does not allow for rendering results through a splatting rendering process.

## C    EXPERIMENTAL DETAILS

### C.1    DETAILS ON DATASETS

We employ the official script from Gaussian Splatting (Kerbl et al., 2023) for colmap to acquire camera poses and the initial point cloud. Consequently, the actual number of images utilized in the

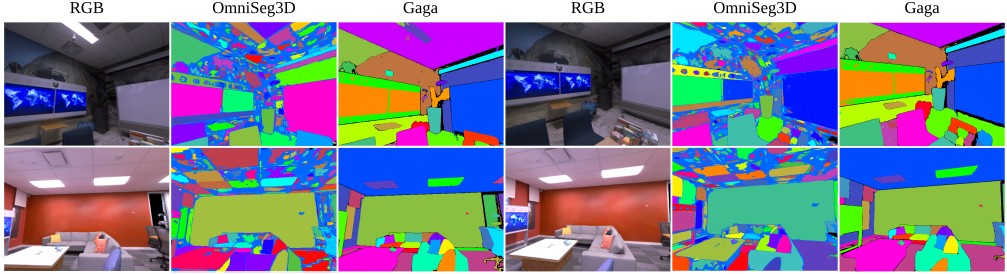

| RGB | OmniSeg3D | Gaga | RGB | OmniSeg3D | Gaga |
|---|---|---|---|---|---|

Figure 17: **Visual comparison with OmniSeg3D.** We use the HDBSCAN algorithm to cluster the rendered features and obtain a global clustering visualization. Compared to *Gaga*, the clustering results from OmniSeg3D lack multi-view consistency.

Table 5: **Selected scenes in Replica and ScanNet datasets.** We select 8 scenes from the Replica dataset following (Zhi et al., 2021), and 7 scenes from the ScanNet dataset following (Wang et al., 2023).

| Dataset | Scene Name | | | |
|---------|------------|------------|------------|------------|
| Replica | office 0 
 office 4 | office 1 
 room 0 | office 2 
 room 1 | office 3 
 room 2 |
| ScanNet | scene 0010_00 
 scene 0088_00 | scene 0012_00 
 scene 0113_00 | scene 0033_00 
 scene 0192_00 | scene 0038_00 |

experiment might be lower than expected due to colmap process failures. Please refer to Tab. 5 for the scene names used in the Replica and ScanNet datasets.

**LERF-Mask Dataset (Ye et al., 2024).** LERF-Mask is based on the LERF dataset (Kerr et al., 2023) and annotated with tasks and ground truth by the author of (Ye et al., 2024). It contains 3 scenes: figurines, ramen, and teatime. For each scene, 6-10 objects are selected as text queries, and Grounding DINO (Liu et al., 2023a) is utilized to select the mask ID from the rendered segmentation.

**Replica Dataset (Straub et al., 2019).** We select 8 scenes from the entire Replica Dataset the same as (Zhi et al., 2021). We use the rendered results provided by authors of (Zhi et al., 2021) and follow their data processing process: for each scene, we uniformly select 20% images as training data and 20% images as test data from all rendered RGB images. This results in 180 training images and 180 test images for each scene.

**Sparsely Sampled Replica Dataset.** For the same 8 scenes as the previous experiment, we randomly sample 30%, 20%, 10%, and 5% of the total 180 training images, resulting in 54, 36, 18, and 9 training images for each task, respectively. The number of test images remains at 180.

**ScanNet Dataset (Dai et al., 2017).** DM-NeRF (Wang et al., 2023) selects 8 scenes from the entire ScanNet dataset. Each scene has approximately 300 images for training and about 100 images for testing. We utilize 7 out of the 8 scenes, excluding "scene 0024_00" due to the subpar 3D reconstruction results in both Gaussian Splatting (Kerbl et al., 2023) and Gaussian Grouping (Ye et al., 2024).

**MipNeRF 360 Dataset (Barron et al., 2021).** We downsample the images by a factor of 4, consistent with the setting in (Ye et al., 2024), to accommodate the large size of the original images. For novel view synthesis evaluation, we set the sample step at 8, the same as the setting in (Kerbl et al., 2023).

## C.2 Details on Evaluation Metrics

Given the disparate mask label assignments between the ground truth segmentation and the predicted segmentation for 3D objects, we find the best linear assignment between the labels based on IoU for quantitative evaluation. Subsequently, we employ IoU $> 0.5$ as the criterion for precision and recall calculations. We outline the pseudocode for the evaluation procedure in Algorithm 1. Note that all annotated segmentation masks are unavailable during training and are only accessible during evaluation as ground truth.

## C.3 Further Implementation Details

For training vanilla 3D Gaussians, we maintain the same parameter setting as (Kerbl et al., 2023). To train the identity encoding, we freeze all the other attributes of Gaussians and use the same parameter setting as (Ye et al., 2024). The identity encoding has 16 dimensions, and the rendered 2D identity encoding is in the shape of $16 \times h \times w$, where $h$ and $w$ represent the height and width of the image. The classifier for predicting mask ID given the 2D identity encoding and selecting Gaussians for editing given the 3D identity encoding shares the same architecture, with 16 input channels. The number of output channels equals the number of groups in the 3D-aware memory bank after associating all images. All experiments are conducted on a single NVIDIA RTX 6000 Ada GPU.

918
919
920
921
922
923
924
925
926
927
928
929
930
931
932
933
934
935
936
937
938
939
940
941
942
943
944
945
946
947
948
949
950
951
952
953
954
955
956
957
958
959
960
961
962
963
964
965
966
967
968
969
970
971

---

**Algorithm 1** Evaluation Metrics

Input *pred_masks* and *gt_masks* are represented in binary format with shape $(n_{image}, n_{mask}, h, w)$, where $n_{image}$ is the number of test images, $n_{mask}$ is the number of predicted or ground truth masks, $h$, $w$ are the height and width of test images.

We use `scipy.optimize.linear_sum_assignment` to solve the linear assignment problem.

    **Function** `evaluate`(pred_masks, gt_masks)
      **Input**: pred_masks (torch.bool), gt_masks (torch.bool)
      **Output**: iou (torch.float), precision (torch.float), recall (torch.float)

    assert len(gt_masks) == len(pred_masks)
    $n_{image} \leftarrow$ len(gt_masks)
    $n_{pred} \leftarrow$ pred_masks.shape[1]
    $n_{gt} \leftarrow$ gt_masks.shape[1]
    iou_matrix $\leftarrow$ torch.zeros$((n_{gt}, \max(n_{gt}, n_{pred})))$
    **for** $i$ **in** $n_{gt}$ **do**
      **for** $j$ **in** $n_{pred}$ **do**
        iou_list $\leftarrow$ []
        **for** $k$ **in** $n_{image}$ **do**
          iou_list.append(`IoU`(gt_masks[k][i], pred_masks[k][j]))
        **end for**
        iou_matrix[i][j] $\leftarrow$ iou_list.mean()
      **end for**
    **end for**
    $gt\_indices$, $pred\_indices \leftarrow$ `linear_assignment`(iou_matrix)
    paired_iou $\leftarrow$ iou_matrix[$gt\_indices$][$pred\_indices$]
    iou $\leftarrow$ paired_iou.mean()
    $n_{correct} \leftarrow$ torch.sum(paired_iou $> 0.5$)
    precision $\leftarrow \frac{n_{correct}}{n_{pred}}$
    recall $\leftarrow \frac{n_{correct}}{n_{gt}}$
    **return** iou, precision, recall

---

Table 6: **Ablation study on the percentage of front Gaussians.** Results for selecting 10%, 20%, 30%, and 100% of front Gaussians as corresponding Gaussians of a mask are presented below. *Gaga* demonstrates stable performance across varying parameters, showcasing its robustness.

| Perc. Front Gaussians (%) | IoU (%) | Precision (%) | Recall (%) |
|---|---|---|---|
| 10 | 46.42 | 39.57 | 51.54 |
| 20 * | **46.50** | 41.52 | **52.50** |
| 30 | 45.73 | **42.31** | 50.88 |
| 100 | 42.26 | 40.19 | 45.95 |

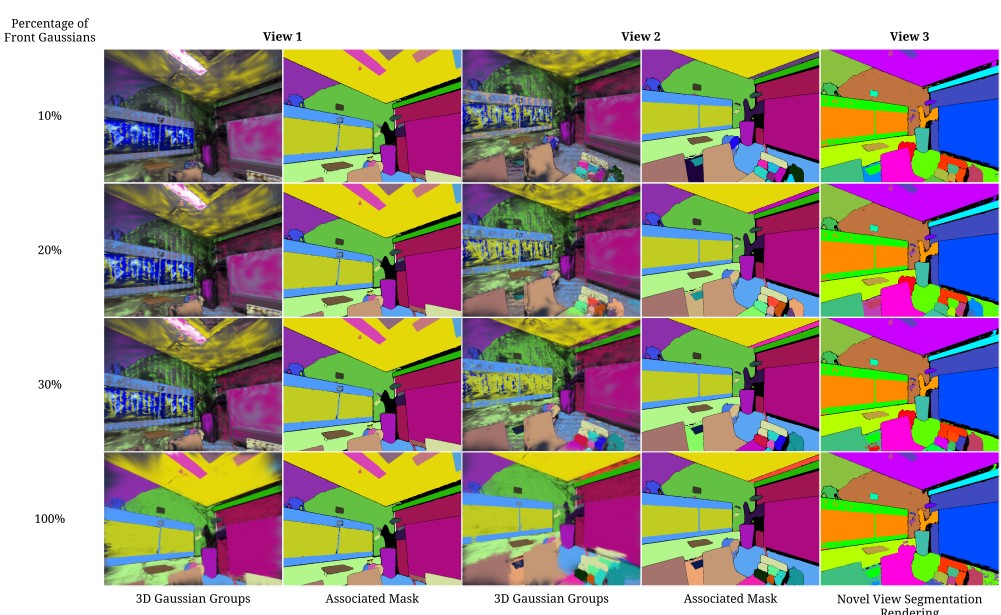

Figure 18: **Visual analysis on percentages of front Gaussians.** When using 10%, 20%, 30% and 100% percentage of front Gaussians, the selected 3D Gaussian Group is more and more dense, meaning more Gaussians are selected as the corresponding Gaussians of a mask. *Gaga* can provide similar segmentation results with all percentages (last column), showcasing its robustness.

## D  SUPPLEMENTARY ABLATION STUDIES

We conduct additional ablation studies on three parameters involved in the process of mask association and find corresponding Gaussians of a mask. These ablation studies are performed on the Replica dataset (Straub et al., 2019), utilizing SAM (Kirillov et al., 2023) as the 2D segmentation model. Parameters denoted with * are used as the default setting. We also provide additional visual comparison results for the mask association methods utilized by Gaussian Grouping (Ye et al., 2024) and *Gaga* in Sec. D.4.

### D.1  PERCENTAGE OF FRONT GAUSSIANS

We present the ablation study on the percentage of front Guassians selected as corresponding Gaussians in Tab. 6. We choose 10%, 20%, 30%, and 100% (*i.e.* selecting all Gaussians splatted to the mask as its corresponding Gaussians) as candidate parameters. The default setting (20%) has a better performance in general. *Gaga* shows stable performance for all candidate parameters, indicating its robustness and it does not rely on cautious parameter selection.

We provide a visual analysis of the percentage of front Gaussians in Fig. 18. As this percentage increases, a greater number of Gaussians are selected as corresponding Gaussians for a mask. Despite this variation, *Gaga* consistently produces similar segmentation maps, demonstrating its robustness to this parameter and showing that the results do not heavily rely on hyperparameter selection.

Table 7: **Ablation study on image partition.** We partition the entire image and its masks into patches to prevent selected corresponding Gaussians from concentrating in a confined region of a mask. Comparison results show that *Gaga* can perform well when the partition process is employed.

| Num. Patches | IoU (%) | Precision (%) | Recall (%) |
|---|---|---|---|
| $1 \times 1$ | 46.08 | 27.88 | 50.67 |
| $16 \times 16$ | 46.11 | 38.22 | 51.62 |
| $32 \times 32$ * | **46.50** | **41.52** | **52.50** |
| $64 \times 64$ | 44.72 | 40.65 | 49.14 |

Table 8: **Ablation study on the overlap threshold.** If the overlap between the current mask and all groups in the memory bank falls below this threshold, we add this mask to the memory bank as a new group. Results indicate that the default setting of 0.1 generally yields better outcomes.

| Overlap Threshold | IoU (%) | Precision (%) | Recall (%) |
|---|---|---|---|
| 0.01 | 43.86 | **44.99** | 48.98 |
| 0.1 * | 46.50 | 41.52 | **52.50** |
| 0.2 | **47.57** | 34.77 | 52.40 |

### D.2 NUMBER OF IMAGE PATCHES DURING PARTITION

We provide the ablation study on the number of image patches used during the image partition process in Tab. 7. Candidate parameters include $1 \times 1$ (without mask partition process), $16 \times 16$, $32 \times 32$, $64 \times 64$. Similar to the results in Tab. 6, *Gaga* remains insensitive to the choice of this parameter as long as the image partition process is in place. Without the mask partition process, there is a significant drop in precision.

### D.3 OVERLAP THRESHOLD

During the group ID assigning process, if none of the existing groups in the memory bank has a larger overlap with the current mask than the threshold, we incorporate this mask into the memory bank as a new group, signifying the discovery of a new 3D object. We present the ablation study on overlap threshold in Tab. 8. When the threshold is set to 0.01, we rarely establish a new group and prefer to associate the mask with an existing group. It provides the best precision but at the expense of inferior IoU performance. Conversely, setting the threshold to 0.2 results in a frequent declaration of new group IDs, yielding the best IoU but a significant decrease in precision. Therefore, we set the threshold to 0.1 to strike a balance in performance across all three metrics.

### D.4 ADDITIONAL COMPARISON ON MASK ASSOCIATION METHODS

We present visual comparison results for two mask association methods, video tracker (Cheng et al., 2023a) utilized by (Ye et al., 2024) and *Gaga*'s 3D-aware memory bank, in Fig. 19. In the "garden" scene of the MipNeRF 360 dataset, the video tracker struggles to track objects in the background, whereas *Gaga* provides associated results for each mask. For the scene in the ScanNet dataset, the video tracker fails to distinguish between four identical sofas, resulting in multiple masks for the same object. Additionally, it assigns different mask IDs to the table in two views. In contrast, *Gaga* precisely locates each object, leading to improved mask association results and better pseudo labels for training segmentation features.

## E LIMITATIONS

Though *Gaga* achieves SOTA performance compared to existing works, there are a few limitations and future works. First, the optimization process of identity encoding and the rest of the Gaussian parameters are independent, this is because we need to first train 3D Gaussians to acquire their spatial location for mask association. While this pipeline allows for the utilization of any pre-trained 3D Gaussians as input without the need to re-train the entire scene, it does require additional training steps. We aim to enable the joint processing of mask association and identity encoding training in future works.

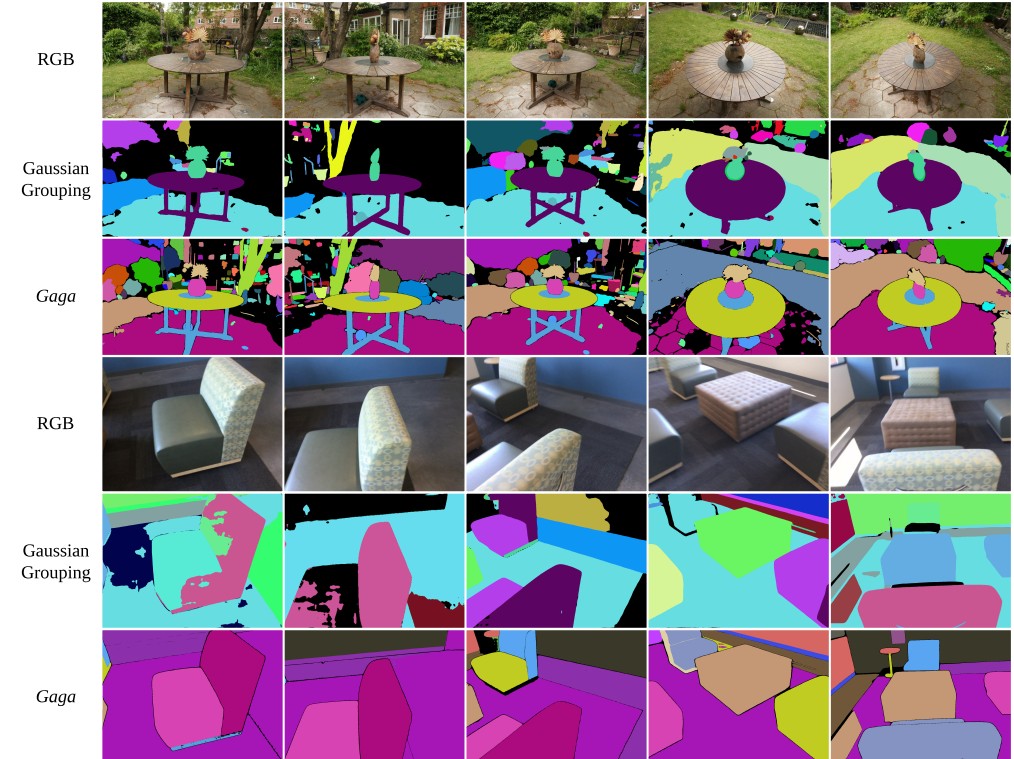

Figure 19: **Visual comparison between different mask association methods.** *Gaga* offers more detailed associated masks, accurately tracks identical objects in the scene and assigns them different mask IDs. Conversely, Gaussian Grouping leaves empty regions in positions where it cannot track masks, and it struggles to provide consistent masks for the same object across views.

Secondly, artifacts may occur in the segmentation rendered by *Gaga* due to inherent inconsistency in the 2D segmentation. For example, an object might be depicted as one mask in the initial view but as two separate masks in subsequent views. This ambiguity introduces challenges to our mask association process. Preprocessing steps such as dividing, merging, or reshaping the 2D segmentation masks could potentially resolve this issue and improve grouping results.

