# OpenReview forum: "Gaga: Group Any Gaussians via 3D-aware Memory Bank"
_ICLR.cc/2025/Conference — ICLR 2025 Conference Withdrawn Submission_

### Official Review · Reviewer_AZ4N · 2024-10-30

**Soundness:** 2
**Presentation:** 3
**Contribution:** 2
**Rating:** 5
**Confidence:** 5

**Summary:**

The paper introduces Gaga, a framework that segments open-world 3D scenes by addressing label inconsistencies in 2D masks. Using a 3D-aware memory bank, Gaga achieves multi-view consistent segmentation by associating 2D masks across views with a universal ID.

**Strengths:**

1. Gaga proposes a novel 3D-aware memory bank that improves multi-view consistency in 3D segmentation by leveraging spatial information from 3D Gaussians, providing an innovative alternative to video-based mask association methods.
2. Extensive experiments validate Gaga's robustness across challenging scenarios, datasets, and limited data.

**Weaknesses:**

1. As noted in Fig. 4a (View 2), the author selects the nearest 20% of Gaussians, considering that the background may contribute to foreground segmentation. However, in cases like Fig. 4 (View 1), where the background does not influence the foreground, this approach may lead to Gaussians from the target partition being omitted.  Please provide visual analysis on how different percentages of nearest Gaussians affect performance across various scenarios.
2.The comparison with existing methods is limited. Including additional comparisons with state-of-the-art approaches, such as Panoptic-Lifting [1] and Contrastive-Lift [2].

[1] Siddiqui Y, Porzi L, Buló S R, et al. Panoptic lifting for 3d scene understanding with neural fields[C]//Proceedings of the IEEE/CVF Conference on Computer Vision and Pattern Recognition. 2023: 9043-9052.
[2] Bhalgat Y, Laina I, Henriques J F, et al. Contrastive lift: 3d object instance segmentation by slow-fast contrastive fusion[J]. arXiv preprint arXiv:2306.04633, 2023.

**Questions:**

1. Could you provide visual examples to demonstrate that selecting the nearest 20% of Gaussians is robust across different scenarios? Additionally, is there any evidence that some Gaussians relevant to the segmentation may be excluded?
2. How your method handles cases where the ground truth segmentation is more fine-grained than their mask partitioning approach allows. For example, in the replica ground truth segmentation, the left and right panels in Figure 4b belong to different masks.  However, the mask partition merges them into a single object, which can reduce segmentation accuracy.

---

> ### Author Response · Authors · 2024-11-22
> **Respnose to Reviewer AZ4N**
>
> We extend our gratitude for the positive and constructive feedback.
>
> We would like to address the raised concerns as follows:
> ___
> > Q1: In cases like Fig. 4 (View 1), where the background does not influence the foreground, this approach may lead to Gaussians from the target partition being omitted. Please provide visual analysis on how different percentages of nearest Gaussians affect performance across various scenarios.**
>
> This phenomenon does happen, and that is why we incorporate the mask partition process to make the selected corresponding Gaussians cover the entire mask uniformly. Comparing row 1 and row 2 in Fig.4 shows the effectiveness of our mask partition strategy. We provide an ablation study on the percentage of corresponding Gaussians in supplementary material Sec. D.1 to show that as long as the mask partition process is in place, our method is robust to the percentage of corresponding Gaussians.
>
> We provide visual analysis on how different percentages of nearest Gaussians in the supplementary material Fig. 15. It shows that the results *Gaga* provided do not heavily rely on hyperparameter selection.
>
> ___
> >Q2: The comparison with existing methods is limited. Including additional comparisons with state-of-the-art approaches, such as Panoptic-Lifting [1] and Contrastive-Lift [2].
>
> Please refer to [Question 2] in the general response.
>
> ___
> >Q3: Could you provide visual examples to demonstrate that selecting the nearest 20% of Gaussians is robust across different scenarios? Additionally, is there any evidence that some Gaussians relevant to the segmentation may be excluded?
>
> For all the visual and numerical experiment results, we use the nearest 20% of Gaussians, so it is robust across different scenarios. As shown in Fig. 4, some Gaussians relevant to the segmentation may be excluded. However, as we set a threshold to decide whether two groups of Gaussians belong to the same 3D object, the excluded Gaussians wouldn't affect the final results.
>
> ___
> > Q4: How your method handles cases where the ground truth segmentation is more fine-grained than their mask partitioning approach allows. For example, in the replica ground truth segmentation, the left and right panels in Figure 4b belong to different masks. However, the mask partition merges them into a single object, which can reduce segmentation accuracy.
>
> Class-agnostic segmentation do not have a ground truth avaiable. The ground truth we utilize for evaluation in Replica and ScanNet datasets is the panoptic segmentation, which is mainly used to demonstrate the advantage of multi-view consistency of our method over previous method, as described in Sec. 4.1 Evaluation Metrics.
>
> The problem you described in Fig.4b happens because, in the first view, SAM segments the left and right panels as one mask. However, in the second view, it segments the two panels as two separate masks. It shows that SAM and other class-agnostic segmentation methods often provide an inconsistent number of masks toward the same object across different views. It is a trade-off whether to keep or abandon small masks, as there is no ground truth for class-agnostic segmentation. *Gaga* does not provide a perfect solution but achieves much better results compared to the previous method. We also describe the limitation of our work in Supp. Sec. 5.
>
> The mask partitioning approach does not merge the masks. In Fig. 4, row 2, it is because SAM segments the left and right panels as one mask, and the mask partition approach helps the selected corresponding Gaussians better cover the entire mask, that's why left and right panels are in one mask. Without the mask partition process, we will see view-inconsistent segmentation results, as shown in row 1.

---

> ### Author Response · Authors · 2024-11-24
>
> We sincerely appreciate your reviews and comments on our paper. Since the discussion phase ends on November 26, we would like to know whether we have addressed all your concerns, such as evidence that selecting the nearest 20% of Gaussians is robust across different scenarios and comparisons with other methods. Please let us know if you have further questions after reading our rebuttal.
>
> We hope to address all the potential issues during the discussion period.
>
> Thank you.

---

> > ### Comment · Reviewer_AZ4N · 2024-11-24
> >
> > Thanks for your efforts. My concerns have been addressed.

---

> > > ### Author Response · Authors · 2024-11-24
> > > **Thank you!**
> > >
> > > Dear Reviewer,
> > >
> > > Thanks for letting us know that we have addressed the concerns of this work. As mentioned in the paper, the source code will be available to the public. This will help the line of multiview consistent segmentation. Since all issues are resolved, could you consider raising the scores?
> > >
> > > Thank you,

---

> > > > ### Author Response · Authors · 2024-11-29
> > > >
> > > > Dear Reviewer,
> > > >
> > > > We sincerely appreciate your review and feedback on our paper. As the discussion period is nearing its end, we want to check if you have any additional concerns or suggestions. If not, we kindly ask if you would consider revising your score. As noted by Reviewer D6rQ, our work demonstrates practical value compared to existing approaches, and we plan to release the code for this work, which we hope will contribute meaningfully to advancing multi-view consistent class-agnostic segmentation.
> > > >
> > > > Thank you!

---

> > > > > ### Author Response · Authors · 2024-12-02
> > > > >
> > > > > Dear Reviewer,
> > > > >
> > > > > As the discussion period ends tomorrow (12/02), we kindly remind you to share any remaining concerns you might have about our paper. If there are no further issues, we would greatly appreciate it if you could consider updating your score. Reviewer D6rQ highlighted the practical value of our work compared to existing methods. Additionally, we intend to release the code for this research, which we hope will play a meaningful role in advancing multi-view consistent class-agnostic segmentation.

---

### Official Review · Reviewer_FmMx · 2024-10-31

**Soundness:** 2
**Presentation:** 3
**Contribution:** 2
**Rating:** 3
**Confidence:** 5

**Summary:**

This paper introduces a new Gaussian splatting segmentation method by designing a mask association algorithm and lifting the created view-consistent mask to the reconstructed Gaussians.

* Technique-wise, to address the inconsistency of 2D masks across different views, this paper designs a 3D-aware memory bank that gathers Gaussians belonging to the same group. This memory bank is utilized to align 2D masks across various views.

* The paper conducts extensive experiments to demonstrate the effectiveness of the proposed method compared to the previous method, i.e., Gaussian grouping using video tracking strategies for mask association.

**Strengths:**

* The proposed method is simple yet effective. It introduces a new mask association algorithm that leverages the intrinsic 3D information of reconstructed Gaussians.
* The paper is well-written and easy to follow.
* The experiments have covered datasets with various settings (e.g., different 2D models, and sparse views).

**Weaknesses:**

* The paper lacks sufficient comparisons with state-of-the-art methods. For example, existing methods have proposed solutions such as the linear assignment [1] or contrastive learning methods [2,3] combined with clustering algorithms to bypass the need for complex mask association preprocessing. These methods should be included as main baselines to verify the effectiveness of the proposed approach.

* Although the mask association is designed to group Gaussians from 2D pose images, the algorithm closely resembles SAM3D[4], which groups point clouds from 2D pose images. Are there any significant differences between the proposed algorithm and SAM3D?  A comprehensive discussion highlighting the technical novelties of the proposed algorithm against existing baselines is needed, if any exist.

* Although this paper introduces a new variant of mask association pre-processing with improved performance, there are no significant differences as it still suffers from similar limitations. Similar to video tracking, the proposed pre-processing involves many hyperparameter tuning and is sensitive to its hyperparameters, such as the percentage for finding corresponding Gaussians, overlap threshold, and image partition, as demonstrated in Tables 1, 2, and 3 in Supp.

* In the proposed algorithm, using a fixed percentage to find corresponding Gaussians is too straightforward, while the optimal percentage should be dependent on the opacity properties. Hence, the preset parameter is likely to be unsuitable for new scenes and may potentially limit the generalization of the proposed method.




[1] Panoptic Lifting for 3D Scene Understanding (CVPR 2023)

[2] Contrastive Lift: 3D Object Instance Segmentation by Slow-Fast Contrastive Fusion. (NeurIPS 2023)

[3] OmniSeg3D: Omniversal 3D Segmentation via Hierarchical Contrastive Learning (CVPR2024).

[4] SAM3D: Segment Anything in 3D Scenes (ICCV'23 Workshop)

**Questions:**

Besides the following question, please refer to the weaknesses section above for my concerns.
* In Supp Figure 1, it is good that the author provides a qualitative comparison with a method using contrastive learning. However, it lacks quantitative comparisons. Besides, the comparison does not make much sense as the GARField model seems not to have converged, and the rendered RGB from GARField is much worse than that from the proposed method. Furthermore, the clustering is sensitive to parameters, but there is no detailed information provided about these settings.
* I strongly encourage the authors to compare their approach with the additional baselines mentioned above and update them in the main experiments, particularly with OmniSeg3D [3], as it provides a 3D-GS implementation, enabling fairer comparisons. To compare with OmniSeg3D, the authors may consider using the same SAM input to train OmniSeg3D, followed by a clustering algorithm for the final results.

---

> ### Author Response · Authors · 2024-11-22
> **Response to Reviewer FmMx Part 1**
>
> We extend our gratitude for the positive and constructive feedback.
>
> We would like to address the raised concerns as follows:
>
> ___
> >Q1: The paper lacks sufficient comparisons with state-of-the-art methods. For example, existing methods have proposed solutions such as the linear assignment [1] or contrastive learning methods [2,3] combined with clustering algorithms to bypass the need for complex mask association preprocessing. These methods should be included as main baselines to verify the effectiveness of the proposed approach.
>
> Please refer to [Question 2] in the general response.
>
> ___
> >Q2: Although the mask association is designed to group Gaussians from 2D pose images, the algorithm closely resembles SAM3D[4], which groups point clouds from 2D pose images. Are there any significant differences between the proposed algorithm and SAM3D? A comprehensive discussion highlighting the technical novelties of the proposed algorithm against existing baselines is needed, if any exist.
>
> SAM3D and our proposed method both use spatial information (point clouds and center of 3D Gaussians) to associate masks from multi-view images. Our method differentiates from SAM3D in the following aspects:
>
> * As SAM3D utilizes point clouds, it cannot provide photo-realistic RGB renderings and watertight segmentation map renderings, and it cannot be used for scene editing. The visual results show that *Gaga* can provide much more detailed segmentation than SAM3D.
>
> * SAM3D obtains point clouds for a frame via depth information and uses "Bidirectional-group-overlap-algorithm" to merge two adjacent point clouds. In Sec. 2.2 in the SAM3D paper, they mention that they compute a correspondence mapping $M$ between point clouds of two adjacent frames $X_1$ and $X_2$, but there are no details on how this computation is done. To get such correspondence mapping, the adjacent frames should have similar camera poses as well as large scene overlaps. Otherwise, it is nontrivial to find the corresponding points between two frames. Also, as mentioned in Sec. 2.3 in the SAM3D paper, a scene in ScanNet consists of hundreds or thousands of images. In contrast, our mask association method uses a camera extrinsic to project 3D Gaussian centers to 2D masks to find corresponding masks of each 3D object and uses a mask partition strategy to make sure the selected corresponding Gaussians cover the mask uniformly. Thus, our method can handle dramatic changes in camera poses and sparsely sampled datasets. On ScanNet, compared to SAM3D which uses hundreds of images to get view-consistent segmentation, our method can obtain meaningful segmentation with as few as nine views, as shown in Sec. 4.4 in the submission and Sec. B.3 in the supplementary, due to the accurate mask association process with camera extrinsic and mask partition strategy.
>
> * SAM3D uses a Bottom-up Merging method to merge the masks of local point clouds into a complete mask. Meanwhile, they need to reduce the number of points in point clouds with grid pooling at the cost of losing finer details. Our method *Gaga* first produces multi-view consistent segmentation masks in 2D, then uses them as pseudo labels to train a segmentation feature (identity encoding) for each 3D Gaussian to enable segmentation rendering, which does not require the complex merging and pooling process in 3D. Meanwhile, training the segmentation features has an advantage in that even if, in a small number of frames, an object is labeled with incorrect segmentation classes, this error can be fixed as more frames have the correct segmentation labels. In other words, our approach is more tolerable for mask association errors, while SAM3D does not have such robustness.

---

> > ### Comment · Reviewer_FmMx · 2024-11-24
> >
> > Thanks for your effort.
> >
> > * For comparisons.
> >
> > The paper originally provides limited comparisons with only one baseline (Gaussian Grouping), as also observed by Reviewer AZ4N. Although it includes the result of OmniSeg-3D during the rebuttal, it lacks sufficient experimental details, and the discussion is confusing.
> >
> >
> > * Discussion on SAM3D.
> >
> > After reading the reply from the author, I do not see the significant differences between the proposed algorithm with SAM3D. For instance, one could simply use the corresponding Gaussian centers as the per-frame point clouds for SAM3D to produce the results. Furthermore, the author should include an additional baseline that utilizes the pre-processing results by SAM3D.

---

> > > ### Author Response · Authors · 2024-11-24
> > >
> > > Thanks for your prompt response. We provide the following additional comments which hopefully can resolve your concerns.
> > >
> > > * Regarding the comparison, we provide comparisons with four baselines on the LERF-Mask dataset: LERF, LEGaussians, LangSplat, and Gaussian Grouping in Sec. 4.2. We would love to provide more experimental details for the comparison with OmniSeg-3D if you have additional questions on that. The experiments on the Replica and ScanNet datasets are designed to test the multi-view consistency of rendered segmentation maps. Our method and Gaussian Grouping are the only two methods we know that provide multi-view consistent class-agnostic segmentation rendering. Methods like GARField and OmniSeg3D can only render multi-view consistent features or per-frame clustering based on feature similarity, which is not multi-view consistent segmentation.
> > >
> > > * Regarding the discussion on SAM3D.
> > >
> > > 1. SAM3D gets per-frame point cloud by mapping the 2D masks to the 3D space according to the depth of each pixel provided by the RGB-D image, as mentioned in their paper Sec. 2.1, which means each point in the point cloud neurally has a correspondence with a 2D map. In our method, finding the corresponding Gaussian centers to a 2D map is one essential step.
> > >
> > > 2. SAM3D uses a bidirectional merging process to find correspondence between the segmentation maps of two adjacent frames. The correspondence comes from the mapping $M$ between the point cloud of frame 1 $X_1$ and the point cloud of frame 1 $X_2$. This mapping does not exist in 3D Gaussians. Meanwhile, this correspondence only exists in adjacent frames, where our method can handle views with large camera pose changes in sparsely sampled datasets.
> > >
> > > 3. Compared to point clouds, 3D Gaussians are much dense; simply projecting a 2D map to 3D Gaussians will include objects from behind, as we discuss in Fig. 4, and design selecting front Gaussians and mask partition to solve. Whereas SAM3D's point clouds are projected according to the depth of each pixel, they do not face such a problem.
> > >
> > > Overall, although the high-level idea of using 3D information for mask association is similar, our method contains many differences in design with SAM3D, due to the difference in per-frame point clouds and 3D Gaussian centers, applicable scenarios, etc. These designs are important contributions to our work and should not be neglected.
> > >
> > > Thank you again for your careful reviews and comments. Please let us know if you have any additional concerns, and we will be happy to address them accordingly.

---

> > > > ### Comment · Reviewer_FmMx · 2024-11-24
> > > >
> > > > > Our method and Gaussian Grouping are the only two methods we know that provide multi-view consistent class-agnostic segmentation rendering. Methods like GARField and OmniSeg3D can only render multi-view consistent features or per-frame clustering based on feature similarity, which is not multi-view consistent segmentation.
> > > >
> > > >
> > > > **I strongly disagree with the claim.**
> > > > The contrastive-based method has already demonstrated its effectiveness in producing multi-view consistent segmentation. Specifically, you can refer to the earlier paper, "Contrastive-Lift," which provides both quantitative and qualitative comparisons for multi-view consistent segmentation. Additionally, methods like GARField and OmniSeg3D also employ contrastive-based approaches. **Why does the author say that these baselines could not provide multi-view consistent segmentation?**
> > > >
> > > > Moreover, panoptic-lifting is an important baseline that should not be overlooked. **These detailed comparisons and thorough discussions are missing.**
> > > >
> > > > > Overall, although the high-level idea of using 3D information for mask association is similar...
> > > >
> > > > **Since the idea is similar, I strongly suggest providing quantitative comparisons to verify if any superiority exists. The comparisons and discussion are crucial to assess the contribution of this work.**

---

> ### Author Response · Authors · 2024-11-22
> **Response to Reviewer FmMx Part 2**
>
> ___
> >Q3: Although this paper introduces a new variant of mask association pre-processing with improved performance, there are no significant differences as it still suffers from similar limitations. Similar to video tracking, the proposed pre-processing involves many hyperparameter tuning and is sensitive to its hyperparameters, such as the percentage for finding corresponding Gaussians, overlap threshold, and image partition, as demonstrated in Tables 1, 2, and 3 in Supp.
>
> The most significant difference of our method is that we utilize spatial information and create a 3D-aware memory bank for mask association, while Gaussian Grouping uses an existing off-the-shelf video tracker to process image frames. We kindly refer the reviewer to check out our supplementary video, which shows that while the camera poses changes, Gaussian Grouping demonstrates significant view inconsistency, while our method does not have such limitations. Similarly, as shown in our sparsely sampled experiments, our method utilizes 3D information and it can handle dramatic changes in camera poses and image-limited scenarios.
>
> For hyperparameter tuning, we use the same set of hyperparameters across all experiments to demonstrate that our method does not depend on specific hyperparameter choices. The ablation study results show minimal variation as the hyperparameters change, highlighting the robustness of our method. This robustness is maintained as long as the core principles behind the hyperparameters—such as selecting front Gaussians to exclude background objects and the mask partitioning process—are adhered to.
>
> ___
> > Q4: In the proposed algorithm, using a fixed percentage to find corresponding Gaussians is too straightforward, while the optimal percentage should be dependent on the opacity properties. Hence, the preset parameter is likely to be unsuitable for new scenes and may potentially limit the generalization of the proposed method.
>
> Our proposed method is straightforward, and we think it is an advance in that we do not need complex algorithm design to incorporate information like the opacity of each Gaussian to perform mask association and we can still achieve good performance. Using opacity could have the following problems:
> * It cannot handle transparent and non-watertight objects.
> * Incorporating the opacity properties will need a pixel-wise process to find the Gaussians projected to each pixel and select Gaussians based on their opacity. It is time consumption. In contrast, our method operates on each 2D mask which is much more efficient.
>
> Furthermore, we experiment with using depth as guidance to find the percentage of Gaussians to be used as corresponding Gaussians and do mask association. The results on the Replica dataset are shown as follows:
>
> | Method | IoU (%) | Precision (%) | Recall (%)|
> | -------- | -------- | -------- | -------- |
> | Depth-guided | 45.73 | 39.42 | 50.92 |
> | *Gaga* | 46.50 | 41.52 | 52.50 |
>
> It shows no improvement than our simple yet effective method and it requires additional depth information.
>
> ___
> >Q5: In Supp Figure 1, it is good that the author provides a qualitative comparison with a method using contrastive learning. However, it lacks quantitative comparisons. Besides, the comparison does not make much sense as the GARField model seems not to have converged, and the rendered RGB from GARField is much worse than that from the proposed method. Furthermore, the clustering is sensitive to parameters, but there is no detailed information provided about these settings.
>
> As GARField is a clustering algorithm, we cannot provide a quantitative comparison since GARField will provide the same clustering ID for masks of different instances. We use the default setting in GARField for training to reconstruct the scene and clustering. The suboptimal RGB rendering results from GARField means that GARField struggles to reconstruct the challenging scenes in Replica dataset or the clustering algorithm effects the NeRF rendering results, which do not happens in our method.
>
> ___
> >Q6: I strongly encourage the authors to compare their approach with the additional baselines mentioned above and update them in the main experiments, particularly with OmniSeg3D [3], as it provides a 3D-GS implementation, enabling fairer comparisons. To compare with OmniSeg3D, the authors may consider using the same SAM input to train OmniSeg3D, followed by a clustering algorithm for the final results.
>
> Please refer to [Question 2] in the general response for comparison with other methods, including OmniSeg3D.

---

> ### Author Response · Authors · 2024-11-24
>
> We sincerely appreciate your reviews and comments on our paper. Since the discussion phase ends on Nov 26, we would like to know whether we have addressed all your concerns, for example, the comparison with OmniSeg3D and hyperparameter sensitivity. Please let us know if you have further questions after reading our rebuttal.
>
> We hope to address all the potential issues during the discussion period.
>
> Thank you.

---

> ### Author Response · Authors · 2024-11-29
>
> We sincerely appreciate your thoughtful feedback and hope the following responses address your concerns.
>
> ___
> > **I strongly disagree with the claim.** The contrastive-based method has already demonstrated its effectiveness in producing multi-view consistent segmentation. Specifically, you can refer to the earlier paper, "Contrastive-Lift," which provides both quantitative and qualitative comparisons for multi-view consistent segmentation. Additionally, methods like GARField and OmniSeg3D also employ contrastive-based approaches. **Why does the author say that these baselines could not provide multi-view consistent segmentation?**
>
> As detailed in our supplementary material (Section B.2), the segmentation maps generated by GARField lack multi-view consistency. Additionally, we provide visual results of rendered global segmentation maps from OmniSeg3D in the supplementary material (Section B.7). Since OmniSeg3D’s segmentation rendering requires manual input of a pixel for single-object segmentation and does not directly produce global segmentation maps, we followed the authors' suggestion and used HDBSCAN to cluster the rendered 2D features to obtain global segmentation. As shown in Fig. 17, the segmentation maps generated by OmniSeg3D also lack multi-view consistency.
>
> Another potential approach to obtain 2D segmentation maps is to first cluster the features of each 3D Gaussian into groups to generate segmentation labels and then render the segmentation labels in 2D. However, this method is actually infeasible since the segmentation labels, being single integers, cannot be directly used in the splatting rendering process. Thus the segmentation labels of overlapping Gaussians cannot be aggregated.
>
> ___
> > Moreover, panoptic-lifting is an important baseline that should not be overlooked. **These detailed comparisons and thorough discussions are missing.**
>
> Panoptic Lifting focuses on 3D close-set panoptic or instance segmentation, which differs from our open-world class-agnostic segmentation. The linear assignment mask association method can be applied to class-agnostic segmentation masks but could not obtain meaningful results, as shown in the following table.
>
> | Method | IoU (%) | Precision (%) | Recall (%)|
> | -------- | -------- | -------- | -------- |
> | Linear Assignment | 1.92 | 1.54 | 0.58 |
> | *Gaga* | 46.50 | 41.52 | 52.50 |
>
> We also provide an additional visual comparison in Supplementary material, Sec. B.6. Similar findings are shown in Gaussian Grouping Fig. 4.
>
> ___
> > **Since the idea is similar, I strongly suggest providing quantitative comparisons to verify if any superiority exists. The comparisons and discussion are crucial to assess the contribution of this work.**
>
> The focus and setting of our method and SAM3D are fundamentally different. Specifically, SAM3D aims to generate a segmented point cloud given multi-view images; this allows them to project segmented pixels to 3D first to obtain segmented point clouds and then modify those projected points (i.e., deleting or merging) in 3D. On the contrary, we focus on taking well-optimized and fixed 3D Gaussians as inputs and predicting segmentation labels for each Gaussian. Our setting is more constrained in case that we cannot delete or change Gaussians, otherwise the well-trained scene will be destroyed and downstream tasks such as editing will become infeasible. To predict segmentation labels under such a restricted setting, we design Gaga that takes each learned 3D Gaussian as it is and leverages inconsistent 2D SAM masks to predict multi-view consistent labels.

---

> > ### Author Response · Authors · 2024-12-02
> >
> > Dear Reviewer,
> >
> > As the discussion period ends tomorrow (12/02), we would like to know if we have addressed all your concerns about our paper. If there are any remaining points or clarifications needed, please let us know, and we will do our best to respond promptly. Otherwise, if you feel that all your concerns have been resolved, we kindly ask you to consider revising your score if appropriate. We greatly appreciate the time and effort you have dedicated to reviewing our work and providing valuable feedback.

---

### Official Review · Reviewer_vYo9 · 2024-11-03

**Soundness:** 3
**Presentation:** 2
**Contribution:** 2
**Rating:** 5
**Confidence:** 3

**Summary:**

The paper introduces Gaga, a framework for reconstructing and segmenting open-world 3D scenes using 2D masks from zero-shot segmentation models. Gaga focuses on achieving consistent label assignments across multiple views, crucial for accurate 3D segmentation. Using a 3D-aware memory bank, it proposes a mask association strategy to unify multi-view masks. Evaluations on datasets like LERF-Mask, Replica, ScanNet, and MipNeRF 360 for tasks such as open-vocabulary 3D queries and 3D segmentation demonstrate promising results, highlighting its potential for downstream scene manipulation applications.

**Strengths:**

1. The introduction of a 3D-aware memory bank reduces dependence on continuous view changes in training images, enhancing
adaptability in segmentation.
2. Comprehensive experimental comparisons demonstrate the proposed method's effectiveness across multiple tasks, validating its versatility.
3. Gaga shows strong robustness to variations in the amount of training data. maintaining high performance even when training images are significantly reduced.

**Weaknesses:**

1. The proposed approach, which uses Gaussian overlap and ID assignment by overlap ratio, is straightforward but highly hand-crafted, relying on manually chosen hyperparameters. While results are promising, the method offers limited innovation to the research community.
2. Although effective on indoor datasets, testing on outdoor or mixed environments would better showcase generalizability.
3. Hyperparameter Sensitivity. Key hyperparameters like percentage $x$, patch size, and overlap ratio are crucial, potentially impacting robustness, especially with distribution shifts.
4. Some sections, such as the Gaussian selection process and Fig.4(b), could be more concise and easier to follow, the figure is not clear to show the problem with pixel-wise startegy.

**Questions:**

1. I am interested in how the performance of Gaga will hold on outdoor datasets and how different hyperparameter settings might impact the results in these scenarios.
2. From the qualitative results in Figures 6 and 7, it appears that Gaga, especially in combination with SAM, consistently produces a black boundary around objects—a phenomenon not seen with other methods. What might be the cause of this effect?

---

> ### Author Response · Authors · 2024-11-22
> **Response to Reviewer vYo9**
>
> We extend our gratitude for the positive and constructive feedback. We would like to address the raised concerns as follows:
> ___
> >Q1: The proposed approach, which uses Gaussian overlap and ID assignment by overlap ratio, is straightforward but highly hand-crafted, relying on manually chosen hyperparameters. While results are promising, the method offers limited innovation to the research community.
>
> Please refer to [Question 1] in the general response.
>
> ---
> >Q2: Although effective on indoor datasets, testing on outdoor or mixed environments would better showcase generalizability. I am interested in how the performance of Gaga will hold on outdoor datasets and how different hyperparameter settings might impact the results in these scenarios.
>
> Thank you for your suggestion. We provide visual results for outdoor scenes on the MipNeRF 360 dataset in the supplementary material, Section B.4. Please refer to the text highlighted in red. These results demonstrate that *Gaga* delivers favorable performance on outdoor scenes compared to Gaussian Grouping.
>
> Additionally, we present *Gaga*'s reconstruction and segmentation results on outdoor scenes (Barn, Horse, and Ignatius) from a large-scale dataset, Tanks and Temples [1]. For this experiment, we used 24 images for scene reconstruction and mask association, with the results detailed in Section B.5 of the supplementary material. It is worth noting that the same set of hyperparameters was applied across this experiment and all others discussed in the paper. The findings highlight *Gaga*'s robustness in diverse testing scenarios, showing that scene-specific parameter tuning is unnecessary for achieving reasonable results.
>
> [1] Knapitsch, Arno, et al. "Tanks and temples: Benchmarking large-scale scene reconstruction." ACM Transactions on Graphics (ToG) 36.4 (2017): 1-13.
> ___
> >Q3: Hyperparameter Sensitivity. Key hyperparameters like percentage, patch size, and overlap ratio are crucial, potentially impacting robustness, especially with distribution shifts.
>
> Please also refer to [Question 1] in the general response.
>
> ---
> > Q4: Some sections, such as the Gaussian selection process and Fig.4(b), could be more concise and easier to follow, the figure is not clear to show the problem with pixel-wise stategy.
>
> Please refer to our supplementary video for an animation illustrating our mask association strategy. For efficiency, our approach operates on patchified masks rather than individual pixels. Additionally, we have updated Fig. 4 (b) for improved clarity; please see the revised text in our updated paper for further details.
>
> ---
> > Q5: Gaga in combination with SAM, consistently produces a black boundary around objects—a phenomenon not seen with other methods.
>
> This is because we directly use the SAM mask after the group ID assignment as pseudo-labels to train the identity encoding. Since the SAM mask includes boundaries, our results also retain these boundaries. In contrast, methods like Gaussian Grouping use pseudo-labels derived from masks processed by DEVA, which do not have boundaries. Consequently, their results also lack boundaries. Furthermore, our method combined with EntitySeg does not show black boundaries as well, as the segmentation maps produced by EntitySeg do not contain boundaries.

---

> > ### Comment · Reviewer_vYo9 · 2024-11-24
> >
> > Thanks for your efforts. My concerns have been addressed.

---

> > > ### Author Response · Authors · 2024-11-24
> > > **Thank you**
> > >
> > > Dear Reviewer,
> > >
> > > Thanks for letting us know that we have addressed the concerns of this work. As mentioned in the paper, the source code will be available to the public. This will help the line of multiview consistent segmentation. Since all issues are resolved, could you consider raising the scores?
> > >
> > > Thank you,

---

> > > ### Author Response · Authors · 2024-11-25
> > >
> > > Dear Reviewer,
> > >
> > > Thank you for your feedback! To further illustrate our method's performance in outdoor scenes and the robustness of its hyperparameters across various scenarios, we have included additional experimental results on the **Tank and Temples** dataset in **Supplementary Material, Section B.5**. Please refer to the highlighted text in red. We hope these results provide a clearer demonstration of our method's generalization capabilities and robustness.

---

> > > > ### Author Response · Authors · 2024-11-29
> > > >
> > > > Dear Reviewer,
> > > >
> > > > We sincerely appreciate your review and feedback on our paper. As the discussion period is nearing its end, we want to check if you have any additional concerns or suggestions. If not, we kindly ask if you would consider revising your score. As noted by Reviewer D6rQ, our work demonstrates practical value compared to existing approaches, and we plan to release the code for this work, which we hope will contribute meaningfully to advancing multi-view consistent class-agnostic segmentation.
> > > >
> > > > Thank you!

---

> > > > > ### Author Response · Authors · 2024-12-02
> > > > >
> > > > > Dear Reviewer,
> > > > >
> > > > > As the discussion period ends tomorrow (12/02), we kindly remind you to share any remaining concerns you might have about our paper. If there are no further issues, we would greatly appreciate it if you could consider updating your score. Reviewer D6rQ highlighted the practical value of our work compared to existing methods. Additionally, we intend to release the code for this research, which we hope will play a meaningful role in advancing multi-view consistent class-agnostic segmentation.

---

> ### Author Response · Authors · 2024-11-24
>
> We sincerely appreciate your reviews and comments on our paper. Since the discussion phase ends on November 26, we would like to know whether we have addressed all your concerns, such as hyperparameter sensitivity and performance on outdoor scenes. Please let us know if you have further questions after reading our rebuttal.
>
> We hope to address all the potential issues during the discussion period.
>
> Thank you.

---

### Official Review · Reviewer_D6rQ · 2024-11-03

**Soundness:** 3
**Presentation:** 3
**Contribution:** 3
**Rating:** 8
**Confidence:** 4

**Summary:**

This paper proposes a new framework called Gaga to segment 3D Gaussians in the open world. The key idea is to assign a multi-view consistent universal mask ID before lifting the 2D result to 3D, which is achieved by searching a memory back for the category ID based on the overlap area with the unprotected mask. Extensive qualitative and quantitative results show the effectiveness of the proposed method.

**Strengths:**

1. Clear motivation and insightful key idea.
2. Simple but effective, supported by extensive experiments on various datasets.
3. A demo video, more extensive details and limitations are provided in the supplementary material.
4. The paper is well-written.

**Weaknesses:**

1. The whole framework is heavily based on the Identity Encoding proposed in Gaussian Grouping by utilizing 3D Gaussian representations. However, in the abstract, it does not mention the use of 3D GS. This should be addressed in the final version.
2. Some important works [1, 2, 3, 4] should be discussed in the final version. Especially for both [1, 2, 4], they all propose to merge 3D results in 3D space, [4] also assign universal mask ID derived from prompt ID for better consistency.

*Refs:*.
[1] SAM3D: Segment Anything in 3D Scenes. ICCV 2023 Workshop.
[2] SAI3D: Segment Any Instance in 3D with Open Vocabularies. CVPR 2024.
[3] OpenNeRF: Open-Set 3D Neural Scene Segmentation with Pixel-Wise Features and Rendered Novel Views. ICLR 2024.
[4] SAMPro3D: Locating SAM Prompts in 3D for Zero-Shot Scene Segmentation. Arxiv 2023.
[5] SAM-guided Graph Cut for 3D Instance Segmentation. ECCV 2024.
[6] Segment3D: Learning Fine-Grained Class-Agnostic 3D Segmentation without Manual Labels. ECCV 2024.

**Justification**:
Since the main framework is heavily based on previous Gaussian Grouping, the idea is incremental. While considering its strong performance and extensive experiments, a Borderline/Weak Accept should be given.

**Questions:**

Can the proposed 3D-aware memory bank be used in other 3D representations such as NeRF, 3D point clouds? Some discussions are expected to be provided.
In the final version, the author may include a brief discussion in their Future Work or Discussion section about potential adaptations or challenges in applying their 3D-aware memory bank to other 3D representations like NeRF or point clouds.

---

> ### Author Response · Authors · 2024-11-22
> **Response to Reviewer D6rQ**
>
> We extend our gratitude for the positive and constructive feedback. We would like to address the raised concerns as follows:
>
> ---
> >Q1: The whole framework is heavily based on the Identity Encoding proposed in Gaussian Grouping by utilizing 3D Gaussian representations. However, in the abstract, it does not mention the use of 3D GS.
>
> The 3D Gaussian representation is employed to reconstruct the scene and provide spatial information for mask association. The abstract has been updated accordingly; please refer to the text highlighted in red.
>
> ---
> >Q2: Some important works [1, 2, 3, 4] should be discussed in the final version. Especially for both [1, 2, 4], they all propose to merge 3D results in 3D space, [4] also assign universal mask ID derived from prompt ID for better consistency.
>
> [1], [2], [4], [5], [6] all focus on point cloud segmentation rather than 3D Gaussians segmentation, and they cannot provide photo-realistic RGB renderings and watertight segmentation map renderings. They also cannot be used for scene editing. [3] focuses on semantic segmentation, which does not face the issue of multi-view inconsistent masks. We will add a section to discuss these works in the revised version. Please also see [Question 2] in the general response for comparison with other methods.
>
> ---
> > Q3: Can the proposed 3D-aware memory bank be used in other 3D representations such as NeRF, 3D point clouds? Some discussions are expected to be provided. In the final version, the author may include a brief discussion in their Future Work or Discussion section about potential adaptations or challenges in applying their 3D-aware memory bank to other 3D representations like NeRF or point clouds.
>
> As we only utilize the center of the 3D Gaussians (x, y, z) to provide spatial information for mask association, 3D point clouds can be utilized in a similar manner. However, point clouds do not offer photo-realistic RGB and water-tight segmentation rendering. NeRF, as an implicit representation, does not provide us explicit spatial information we need, so cannot be directly adopted in our method. We will include this discussion in our revised version.

---

> ### Author Response · Authors · 2024-11-24
>
> We sincerely appreciate your reviews and comments on our paper. Since the discussion phase ends on Nov 26, we would like to know whether we have addressed all your concerns. Please let us know if you have further questions after reading our rebuttal.
>
> We hope to address all the potential issues during the discussion period.
>
> Thank you.

---

> ### Comment · Reviewer_D6rQ · 2024-11-25
> **Would Raise the Score**
>
> After reading the rebuttal from authors and other reviews, I would raise my rating to Accept, towards a positive rating.
>
> The main reason is that the explanations about novelty are still not enough. However, considering the performance, this paper shows certain practical values.
>
> Considering the high requirement of ICLR bar, a Weak Accept would be best.

---

> ### Author Response · Authors · 2024-11-25
> **Thank you**
>
> Thank you for your response and your recognition of our work! We truly appreciate your thorough and valuable feedback, which has greatly helped improve our work. We will ensure the revision and discussion are included in our final version.

---

### Author Response · Authors · 2024-11-22
**General Response to All Reviewers**

We sincerely appreciate your positive and constructive feedback. We would like to address common questions first in the following:

**[Question 1] Hyperparameters sensitivity.**

We use three hyperparameters in our experiments: Percentage of front Gaussians, number of image patches during mask partition, and Gaussian overlap threshold. We use the same set of hyperparameters (20%, 32 $\times$ 32, 0.1) for all experiments in the paper to show that *Gaga* does not require scene-specific hyperparameter tuning for optimal results. Additionally, we provide ablation studies in the supplementary material (Sec. D) to show that our method is robust to hyperparameter selection, as long as the key concepts behind these hyperparameters (i.e., selecting front Gaussians to avoid including background objects, the mask partition process, etc.) are in place. To summarize, *Gaga* is not sensitive to hyperparameters and does not need hand-craft tuning.

Meanwhile, language-embedding-based methods, such as LERF, LEGaussians, LangSplat, and contrastive learning-based approaches like OmniSeg3D, all rely on hyperparameters, such as similarity threshold to determine whether two pixels should be grouped into the same segment. Similarly, Gaussian Grouping employs the video tracking method DEVA, which also requires manually set parameters. As reviewer FmMx noted, GARField also depends on hyperparameter selection and is sensitive to these parameters. Although our method involves hyperparameters, this is a common practice in related research, and our approach is not sensitive to these parameters.

**[Question 2] Comparison with existing methods.**

* **Panoptic Lifting** and **Contrastive Lift**: Both of these two methods focus on 3D panoptic segmentation or instance segmentation. They associate inconsistent labels of instance segmentation masks where class labels are known. Instead, *Gaga* focuses on class-agnostic segmentation. We tried to apply Panoptic Lifting's linear assignment mask association method to class-agnostic segmentation masks but could not obtain meaningful results. Fig. 4 row 1 of the Gaussian Grouping paper also shows similar findings.
* **OmniSeg3D**: We compare with OmniSeg3D on LERF-Mask Dataset and the results are shown as follows. Note that OmniSeg3D requires a manually input pixel x, y to select target segmentation with feature similarity, we use GroundingDINO to get a segmentation mask for the test prompt and select the center of the mask as the manually input pixel x, y. We use similarity threshold 0.9 as there's no official setting in OmniSeg3D, and 0.9 gives us the best results.

*mIoU(%)*

| Method | figurines | ramen | teatime| Overall |
| --- | --- | --- | --- | --- |
| OmniSeg3D | 70.21 | 60.66 | 68.39 | 66.42 |
| *Gaga* | 90.66 | 64.15 | 69.34 | 74.71 |

*mBIoU(%)*

| Method | figurines | ramen | teatime| Overall |
| --- | --- | --- | --- | --- |
| OmniSeg3D | 68.99 | 54.91 | 62.68 | 62.20 |
| *Gaga* | 88.99 | 61.59 | 65.99 | 72.19 |

Furthermore, OmniSeg3D (similarly GARField) attaches features learned through contrastive learning to each Gaussian, which is different from our learned identity encoding. Their features cannot be used for rendering multi-view consistent segmentation maps, which is the focus of our approach. This also explains why we cannot compare with their methods on Replica or ScanNet datasets, as their segmentation maps are not view-consistent.

* **SAM3D**: SAM3D focuses on point clouds segmentation. Please refer to the response to reviewer FmMx for a detailed discussion about the difference between SAM3D and *Gaga*.

---

> ### Comment · Reviewer_FmMx · 2024-11-23
>
> First, thanks for your response. But I still have concerns after carefully reading your reply.
>
> * [Question 1] Hyperparameters sensitivity.
>
> I still think the proposed method is sensitive to hyper-parameters. One key reason for the sensitivity to hyper-parameters is that the pre-processing approach of the proposed method is based on the Euclidean space, thus the hyper-parameters are closely related to varying physical sizes for objects across different scenes.
>
> * [Question 2] Comparison with existing methods.
>
>
> The discussion about the comparisons with OmniSeg3D seems to be inaccurate.
> > OmniSeg3D (similarly GARField) attaches features learned through contrastive learning to each Gaussian, which is different from our learned identity encoding.
>
> Specifically, both OmniSeg3D and the proposed method use learned identity encoding, as confirmed after checking the official implementation of OmniSeg3D.  https://github.com/OceanYing/OmniSeg3D-GS
>
> >  Their features cannot be used for rendering multi-view consistent segmentation maps, which is the focus of our approach.
>
> The features from OminiSeg3D also can be used for rendering multi-view consistent segmentation maps.
>
>
> These confusions raise further concerns about whether the above comparison is conducted in a fair setting.

---

> > ### Author Response · Authors · 2024-11-23
> >
> > Thank you for carefully reviewing our response and providing feedback. We would like to address your concern as follows:
> > ___
> > > I still think the proposed method is sensitive to hyper-parameters. One key reason for the sensitivity to hyper-parameters is that the pre-processing approach of the proposed method is based on the Euclidean space, thus the hyper-parameters are closely related to varying physical sizes for objects across different scenes.
> >
> > It's true that we will encounter different physical sizes for objects across different scenes. The same set of hyper-parameters can be used to handle all object sizes because we use the mask partition strategy, as introduced in Fig. 4 (b) to divide the big objects/masks into small patches. We find this strategy effective as it makes sure the selected Gaussians can cover the entire object even if the object has a large physical size.
> >
> > ___
> > > The discussion about the comparisons with OmniSeg3D seems to be inaccurate.
> >
> > You can find in render_omni.py that the authors of OmniSeg3D commented out the parts that require a classifier, for example, the following line:
> > https://github.com/OceanYing/OmniSeg3D-GS/blob/99eb67295ee14be4f451875eff05add3f47c42b1/render_omni.py#L97
> >
> > Meanwhile, all results shown in the OmniSeg3D paper are multi-view consistent feature maps and segmented 3D objects. There are no visual results for multi-view consistent segmentation maps, where pixels of the same 3D objects across views are represented with the same color.
> >
> > They adopt codes from Gaussian Grouping to implement their GS version, which is why they also train a 16-dimension feature, similar to Gaussian Grouping and our work. However, their feature is learned through contrastive learning; their feature captures the semantic difference between pixels, not the segmentation label ID, as they do not co-train a classifier to classify the feature into segmentation labels as Gaussian Grouping and our work. Given a pixel, they can find pixels with high feature similarities and provide a segmentation map for the selected object. They can also do clustering to get a segmentation map, as shown in Fig. 6 in their paper, but there's no guarantee that the same cluster ID will be assigned to the same 3D object for the segmentation map in a different view. They discuss in section 5, limitation, that
> > >Besides, since the contrastive learning is applied on single images, two objects that have never appeared in the same image may have similar semantic feature.
> >
> > It further confirms that the feature they learn is not identity encoding, as identity encoding presents a unique segmentation label ID for all objects in a scene, which we achieve by keeping a memory bank to store all processed objects.
> >
> > We hope the above comments can address your concern, and please let us know if you have further comments!

---

> ### Comment · Reviewer_FmMx · 2024-11-24
>
> > [Question 1] Hyperparameters sensitivity.
>
> As mentioned previously, **"Hyperparameter sensitivity" is an inherent issue of this work due to its sole reliance on "Euclidean feature space"**. Although this paper proposes various strategies to handle this issue, the hand-crafted design will introduce more sensitivity.
> > identity encoding
>
> Technique-wise, "feature" and "identity encoding" refer to the **same concept**, but the author's choice of **different names** is confusing!
>
> > Effectiveness of the contrastive-based method: the feature they learn is not identity encoding, as identity encoding presents a unique segmentation label ID for all objects in a scene...
>
> **The contrastive-based method has already demonstrated its effectiveness in producing multi-view consistent segmentation.** Specifically, you can refer to the earlier paper, "Contrastive-Lift," which provides both quantitative and qualitative comparisons for multi-view consistent segmentation. Additionally, methods like GARField and OmniSeg3D also employ contrastive-based approaches. **Why does the author say that these baselines could not provide multi-view consistent segmentation?**

---

> > ### Author Response · Authors · 2024-12-01
> >
> > We sincerely appreciate your thoughtful feedback and hope that the following responses address your concerns.
> > ___
> > > As mentioned previously, "Hyperparameter sensitivity" is an inherent issue of this work due to its sole reliance on "Euclidean feature space." Although this paper proposes various strategies to handle this issue, the hand-crafted design will introduce more sensitivity.
> >
> > Could you kindly elaborate on the concept of "Euclidean feature space" and its application in our paper? Additionally, could you please explain in more detail why our proposed design is expected to introduce greater sensitivity?
> >
> > ___
> > > Technique-wise, "feature" and "identity encoding" refer to the same concept, but the author's choice of different names is confusing!
> >
> > As discussed in our paper (Section 3.1) and in the Gaussian Grouping section (Section 3.2.C), each identity encoding uniquely represents each 3D object's instance ID within the entire scene. This encoding can be classified using a co-trained classifier to generate segmentation labels for all 3D instances **in the entire scene**. In contrast, the contrastive learned features in OmniSeg3D and GARField do not encode the instance IDs of objects. Instead, they represent the similarity between pixels **within a single image**. To group pixels, one must calculate their similarity to form clusters. Meanwhile, these features are not trained to distinguish objects appearing in different images, as noted in the limitations section of OmniSeg3D (Section 5).
> >
> > ___
> > > The contrastive-based method has already demonstrated its effectiveness in producing multi-view consistent segmentation. Specifically, you can refer to the earlier paper, "Contrastive-Lift," which provides both quantitative and qualitative comparisons for multi-view consistent segmentation. Additionally, methods like GARField and OmniSeg3D also employ contrastive-based approaches. Why does the author say that these baselines could not provide multi-view consistent segmentation?
> >
> > As detailed in our supplementary material (Section B.2), the segmentation maps generated by GARField lack multi-view consistency. Additionally, we provide visual results of rendered global segmentation maps from OmniSeg3D in supplementary material (Section B.7). Since OmniSeg3D’s segmentation rendering requires manual input of a pixel for single-object segmentation and does not directly produce global segmentation maps, we followed the authors' suggestion and used HDBSCAN to cluster the rendered 2D features to obtain global segmentation. As shown in Fig. 17, the segmentation maps generated by OmniSeg3D also lack multi-view consistency.
> >
> > Another potential approach to obtain 2D segmentation maps is to first cluster the features of each 3D Gaussian into groups to generate segmentation labels and then render the segmentation labels in 2D. However, this method is actually infeasible since the segmentation labels, being single integers, cannot be directly used in the splatting rendering process. Thus the segmentation labels of overlapping Gaussians cannot be simply aggregated.

---

### Comment · Reviewer_FmMx · 2024-11-24
**Severe issues of this paper**

After reading the initial submission and the rebuttal, I found that the following severe issues have not been resolved in this paper:


* **Comparisons seem spurious to me.**

During the rebuttal, the author provides the results of OmniSeg-3D on the LERF-Masked dataset. However, the provided result (mIoU: 66.4) of OmniSeg-3D is not aligned with the results (mIoU: 79.4) reported in the recent literature [1]. For me, this discrepancy is unusual, and the author should provide more clarification for this.

[1] Click-Gaussian: Interactive Segmentation to Any 3D Gaussians. ECCV 2024.

Besides, the author provides qualitative comparisons with the one NeRF-based baseline (i.e., GARField) in the supplementary materials, but it seems that the NeRF backbone has not converged sufficiently to reconstruct the 3D scene. Consequently, it does not make much sense to compare the segmentation results as the quality of segmentation is highly dependent on the quality of reconstruction. Furthermore, the author should provide more details about the experiment settings, as the results presented do not align with typical experimental outcomes on the widely-used Replica dataset.

* **The originality of the proposed idea is not clear.**

The originality of the proposed idea (mask association algorithm) is not clear. Essentially, this algorithm is similar to point cloud segmentation algorithms like SAM3D. It is important to note that the proposed algorithm only uses the Gaussian mean (3D point) as the algorithm input. However, I **did not** observe any significant differences compared to SAM3D, and **no** quantitative comparisons were provided. **Given the above issue, I have a strong concern about whether the paper merely adapts the SAM3D algorithm to solve the mask association algorithm.** If so, the current version does not meet the standards expected of an ICLR paper.


* **Lack of comparisons with existing baselines.**

For the main baseline for multi-view consistent segmentation, this paper only compares it with Gaussian Grouping, which is far from sufficient. After I highlighted that there are many important baselines including Panoptic Lifting, Contrastive Lift, and OmniSeg-3D, the author provides the results of **only OmniSeg-3D** on a **single** dataset (LERF-Masked). Although the results of OmniSeg-3D are lower than those of the proposed method on this dataset, I do not see **any convincing reasons** for this outcome. Particularly, some discussions are confusing and even inaccurate. These confusions raise further concerns about whether the newly added comparison is conducted in a fair setting.

By the way, I noted that the author provides some comparisons with language-based (i.e., CLIP) methods. However, the results do not make much sense as these methods primarily focus on encoding open-vocabulary features and do not concentrate on providing an accurate segmentation mask output.

* **Hyperparameters sensitivity.**

Since the proposed association algorithm is a **purely hand-crafted** algorithm based on Euclidean space, the hyperparameters are closely related to varying physical sizes of objects across different scenes. Although the proposed algorithm provided various strategies to solve this problem, they resemble engineering tricks more than robust solutions. Given that the single algorithm contains too many hyperparameters (percentage for finding corresponding Gaussians, overlap threshold, and image partition), it is difficult to verify its robustness through experiments on limited datasets. Hence, I have concerns about the sensitivity to hyperparameters.

---

> ### Author Response · Authors · 2024-11-25
>
> We appreciate your feedback and would like to address your concern as follows.
> ___
> > Comparisons seem spurious to me.
>
> The reason for results by OmniSeg3D in this paper differ from the one Click-Gaussian provided is because OmniSeg3D and Click-Gaussian require **a manually input pixel position to select the target object and mask**. Instead in our experiment setting, the target objects are selected automatically by GroundingDINO, given a text prompt. This setting aligns with the experimental setting in Gaussian Grouping, and it is unfair to compare their results obtained by manually selecting 3D objects with our results obtained by selecting 3D objects with GroundingDINO.
>
> For the comparison with GARField, we use their default setting to train the scene and get segmentation results. The results also verify that CARField does not provide consistent segmentation results for multi-views but only clustering results for a single view.
>
> About the experiment on the Replica dataset. The Experiments in Sec. 4.3 use the evaluation metric detailed in Supp. Alg. 1. Gaga targets 3D open-world class-agnostic segmentation, which does not have ground truth. Hence, we adopt the ground truth panoptic segmentation masks as references. Since the class-agnostic segmentation often provides more detailed masks, it is unfair to directly compare Gaga's results with typical experimental outcomes of methods for close-set panoptic segmentation i.e., Panoptic Lifting, DM-NeRF with a panoptic segmentation backbone. Therefore, we add a baseline SAM (Upper Bound) on Replica, using SAM to process rendered RGBs from Gaussian Splatting and evaluate on each single frame without considering multi-view consistency. This baseline can serve as an upper bound to show the inherent difference between class-agnostic and panoptic segmentation.
>
> | Method | IoU (%) | Precision (%) | Recall (%)|
> | -------- | -------- | -------- | -------- |
> | SAM (Upper Bound) | 60.89 | 57.07 | 67.16 |
> | *Gaga* | 46.50 | 41.52 | 52.50 |
>
> ---
> > The originality of the proposed idea is not clear.
>
> **Differentiation from SAM3D**. The focus and setting of our method and SAM3D are fundamentally different. Specifically, SAM3D aims to generate a semantic point cloud given multi-view images, this allows them to project pixels to 3D first and then modify those projected points (i.e., deleting or merging) in 3D. On the contrary, we focus on taking well-optimized and fixed 3D Gaussians as inputs and predicting segmentation labels for each Gaussian. Our setting is more constrained in case that we cannot delete or change Gaussians, otherwise the well-trained scene will be destroyed and downstream tasks such as editing will become infeasible. To predict segmentation labels under such a restricted setting, we design *Gaga* that takes each learned 3D Gaussian as it is and leverages inconsistent 2D SAM masks to predict multi-view consistent labels.
>
> ___
> > Lack of comparisons with existing baselines.
>
> 1. *Comparison with Paonoptic Lifting and Contrastive Lift.*
> As we state in the pervious rebuttal, Panoptic Lifting and Contrastive Lift focus on 3D close-set panoptic or instance segmentation, which differs from our open-world class-agnostic segmentation. The linear assignment mask association method can be applied to class-agnostic segmentation masks but could not obtain meaningful results, as shown in the following table.
>
> | Method | IoU (%) | Precision (%) | Recall (%)|
> | -------- | -------- | -------- | -------- |
> | Linear Assignment | 1.92 | 1.54 | 0.58 |
> | *Gaga* | 46.50 | 41.52 | 52.50 |
>
> We also provide an additional visual comparison in Supplementary material, Sec. B.6. Similar findings are shown in Gaussian Grouping Fig. 4.
>
> 2. *Comparison with OmniSeg3D and GARField.*
> The difference between their contrastive learned feature and identity encoding. As introduced in our paper, Sec. 3.1 and in Gaussian Grouping Sec. 3.2.C, the identity represents its instance ID of the scene and can be classified via a co-trained classifier and produce segmentation labels for all the instances of the scene. The contrastive learned feature in OmniSeg3D and GARField does not represent the instance ID of objects. One needs to calculate the similarity between pixels to get a cluster that groups pixels together. As shown in our supplementary material B.2, the segmentation maps get through GARField are not multi-view consistent. In the codes of OmniSeg3D (https://github.com/OceanYing/OmniSeg3D-GS/blob/99eb67295ee14be4f451875eff05add3f47c42b1/render_omni_gui.py#L516), it shows that OmniSeg3D only provides codes for segment one 3D object at a time, which differs from the global segment anything maps our method can Gaussian Grouping offer.

---

> ### Author Response · Authors · 2024-11-25
>
> ___
> > "Hyperparameter sensitivity" is an inherent issue of this work due to its sole reliance on "Euclidean feature space"
>
> Could you please elaborate on "Euclidean feature space" and where we used it in the paper. Meanwhile I respectfully disagree with the claim that we experimented on limited datasets, as 4 different datasets as well as sparsely sampled Replica dataset are utilized in our experiments, each with 3 to 8 scenes, all with the same set of hyperparameters. OmniSeg3D uses 1 dataset (Replica), GARField uses 1 dataset (LERF).

---

> ### Author Response · Authors · 2024-11-25
> **Additional Comparison with OmniSeg3D**
>
> In addition, we provide visual results of rendered global segmentation maps by OmniSeg3D in supplementary material Sec. B.7. As the segmentation rendering of OmniSeg3D requires a manually input pixel for single object segmentation and does not generate global segmentation maps, we use the HDBSCAN to cluster the rendered 2D features to get global segmentation. As shown in Fig. 17, the segmentation maps produced by OmniSeg3D **do not** have multi-view consistency. Another possible way to obtain 2D segmentation maps is to first cluster features of each 3D Gaussian into groups and then render the segmentation labels in 2D. However, the obtained group ID cannot be directly utilized for the splatting rendering process since it is a single integer, thus the segmentation labels of overlapping Gaussians cannot be simply aggregated. If we change the group ID into one hot encoding format, which enables splatting rendering, a 16-dimension vector can only represent 16 groups, while the identity encoding used in *Gaga* can represent hundreds of group IDs.

---

### Comment · Reviewer_D6rQ · 2024-11-25
**I Believe the Practical Value of This Paper Deserves a Borderline Accept or Weak Accept**

Dear All Reviewer,

I have carefully read all of your reviews.

I want to raise one key point is that: this paper indeed shows some practical values compared to previous works.

Although there may be some minor issues or limited theoretical novelty of this paper, the practical value of it may still contribute a lot to the community. Also, the experiments are very extensive, from my understanding.

As for **the comparison with SAM3D** as mentioned by Reviewer FmMx, from my understanding, the proposed method in this submission solves the multi-view consistency problem, while SAM3D can not as it uses automatic-SAM on each frame (which must cause the view-inconsistency problem). Considering this, **the novelty of the submission is valid**. (However, the authors are still encouraged to provide more discussions with SAM3D, SAMPro3D, etc.)

I think that the standard to judge a paper should not be single or fixed.

I have read lots of accepted papers of recent ICLR papers or other top conferences, and I believe that this paper deserves a Borderline Accept or even Weak Accept.

Currently, to defend my views, I would raise the score to Accept (since there is no Weak Accept option)

---

### Note · Authors · 2025-03-01

I have read and agree with the venue's withdrawal policy on behalf of myself and my co-authors.

---

### Meta-Review · Area_Chair_GjnW · 2024-12-14

**Metareview:**

The paper proposes an approach to multi-view consistent grouping via gaussian splatting. It builds on the recent Gaussian grouping work and replaces the video-tracker-based mask assignment across frames with an overlap-based assignment and a memory bank to track group assignments. Reviewers found the assignment novel, technically sound, and well explained. Extensive qualitative and quantitative results demonstrate improvements over the baseline.

Reviewers remarked a missing discussion of and comparison to relevant related work. Some reviewers considered the contributions to be incremental and insufficient for acceptance, which was countered by the strong performance delta to the baseline. Reviewers were further concerned about the sensitivity to hyperparameter choices that they expected to be domain-dependent (e.g. outdoor scenes requiring a different threshold to cut off Gaussians than object-centric scenes.).

The AC agrees with reviewers that the technical contribution is rather incremental, but the significantly better results in comparison to the baseline could warrant acceptance. This would however require an experimental evaluation not only against the baseline approach, but other methods representative of the current state of the art to signify that the proposed change is meaningful not only with respect to the baseline but the current state of the art. Experimental results added during the discussion phase together with statements made by the authors raised doubts over the thoroughness of the baseline setup and thus the fairness of evalution. With these doubts not cleared by the authors during the discussion, the AC can only recommend to reject the paper.

**Additional Comments On Reviewer Discussion:**

The discussion evolved around experiments on hyperparameters, and comparisons to additional baselines as these were major concerns. The authors added experimental results to the discussion, comparing the method to other prior work and further results on the hyperparameters.
Unfortunately, the added experimental results raised several concerns about the fairness of the evaluation:

- Numbers reported for the same method and task significantly differ to those numbers from concurrent work (to the favour of the proposed approach).
- Qualitative results and descriptions by the authors implied that the baselines were not run in the correct settings. For instance, visual results from GARfield showed incomplete convergence and GARfield was reported to cluster in 2D, not producing multi-view consistent masks. This is one of two possible ways of clustering the GARfield paper describes, and, unfortunately, the wrong one for this experiment (as the other is in 3D, producing renderable 3D point clouds which would yield multi-view consistent 2D masks).
- No results were added to the main tables of the paper during revision, but only to the appendix.
- Reviewers requested comparisons to relevant related work at multiple occasions. Instead of adding the comparisons, authors argued for the related work being different, which missed the point.

---

### Decision · Program_Chairs · 2025-01-22

Reject